Open Access EGU

# spyro: a Firedrake-based wave propagation and full waveform inversion finite element solver

Keith J. Roberts[1], Alexandre Olender[2], Lucas Franceschini[2], Robert C. Kirby[3], Rafael S. Gioria[1], and Bruno S. Carmo[2]

[1]Dept. of Mining and Petroleum Engineering, Escola Politécnica, University of São Paulo
[2]Dept. of Mechanical Engineering, Escola Politécnica, University of São Paulo
[3]Dept. of Mathematics, Baylor University

**Correspondence:** Keith J. Roberts (keithrbt0@gmail.com)

**Abstract.** In this article, we introduce spyro, a software stack to solve acoustic wave propagation in heterogeneous domains and perform full waveform inversion (FWI) employing the finite element framework from Firedrake, a high-level Python package for the automated solution of partial differential equations using the finite element method. The capability of the software is demonstrated by using a continuous Galerkin approach to perform FWI for seismic velocity model building, considering realistic geophysics examples. A time-domain FWI approach is detailed that uses meshes composed of variably sized triangular elements to discretize the domain. To resolve both the forward and adjoint-state equations, and to calculate a mesh-independent gradient associated with the FWI process, a fully-explicit, variable higher-order (up to degree $k = 5$ in 2D and $k = 3$ in 3D) mass lumping method is used. We show that, by adapting the triangular elements to the expected peak source frequency and properties of the wavefield (e.g., local P-wavespeed) and by leveraging higher-order basis functions, the number of degrees-of-freedom necessary to discretize the domain can be reduced. Results from wave simulations and FWIs in both 2D and 3D highlight our developments and demonstrate the benefits and challenges with using triangular meshes adapted to the material properties.

## 1 Introduction

The construction of models consistent with observations of Earth's physical properties can be posed mathematically as solving an inverse problem referred to as full waveform inversion (FWI) (Lines and Newrick, 2004; Virieux and Operto, 2009; Fichtner, 2011; Brittan et al., 2013). FWI is used extensively in geophysical exploration studies in the search for raw materials such as oil and gas (Gras et al., 2019; Fruehn et al., 2019). The attraction of the FWI approach is the promise of deriving higher fidelity models from acquired seismic data as compared to other less complex and less costly methods (e.g., time travel tomography) (Lines and Newrick, 2004). However, the FWI problem is challenging to apply in practice since there exists a non-unique configuration of data that can best explain the observations. Besides this, the associated computational cost to simulate wave propagation in expansive 2D and 3D domains can quickly become extremely demanding.

The basic method of FWI requires several computationally and memory expensive components that need to be executed iteratively potentially dozens of times to arrive at an optimized model (Virieux and Operto, 2009; Fichtner, 2011; Pratt and

Worthington, 1990; Bunks et al., 1995; Jones, 2019; Basker et al., 2016). Each iteration of FWI requires the simulation of acoustic or elastic waves in an arbitrarily heterogeneous medium, which can only be accomplished via numerical approaches. Further, in order to sufficiently illuminate a given domain and provide sufficient information to produce a solution to the inverse problem, many wave simulations are often required. As a result, the primary computational expense of the FWI scales with the cost to numerically simulate wave propagation. Thus, by more efficiently modeling wave propagation, the process of FWI can be accelerated.

Considering the computational cost of solving the wave equation is important to efficiently performing FWI, finite difference methods are often used to model wave propagation. Finite difference methods are well-studied in the context of seismic application in part because they can be highly optimized for computational performance especially so with the help of recent packages such as Devito (Louboutin et al., 2019; Witte et al., 2019). However, canonical finite difference methods use structured grids to represent the domain and inefficiently represent irregular geometries and/or large regional/global domains without the use of more sophisticated methods (e.g., Liu et al., 2008). Consequently for these cases, approaches such as finite element methods (FEM) are often preferred as they discretize the domain with an unstructured mesh of, most commonly, variable sized quadrilaterals/hexahedrals or triangles/tetrahedrals (e.g., Krischer et al., 2015; Modrak et al., 2018; Zhang, 2019; Peter et al., 2011; Anquez et al., 2019; van Driel et al., 2020; Thrastarson et al., 2020; Trinh et al., 2019). The element size can be adapted to the variation of the local shortest wavelength when the seismic velocity field is spatially variable (e.g., Etienne et al., 2009) or to the source location (e.g., van Driel et al., 2020; Thrastarson et al., 2020) to reduce the number of degrees-of-freedom (DoF). For this reason in part, Spectral Element Methods (SEM) using tensor-based quadrilaterals/hexahedrals are widely used in geophysical applications for expansive regional and global domains (Modrak et al., 2018; Fichtner, 2011; Lyu et al., 2020; Fathi et al., 2015; Patera, 1984; Seriani and Priolo, 1994). Furthermore, since the stability condition for explicit time-marching schemes depends on the maximal local ratio of velocity to mesh size, local mesh size adaptation can decrease the overall work-load associated with the wave propagation.

Despite the advantage unstructured meshes appear to offer to FWI there are several major difficulties associated with using them that we attempt to address in this work. 1) The computational burden associated with solving a sparse system of equations arising from the discretization with finite elements, 2) the generation and distribution of variable resolution unstructured meshes 3) code complexity and optimization associated with programming finite element methods themselves. Unlike in the case of SEM, in which the domain is discretized using tensor-based hexahedral elements that result in diagonal mass matrices (e.g., mass lumped) and can be efficiently time marched (Peter et al., 2011; Patera, 1984), standard conforming simplex finite elements produce a large sparse system of equations, even for explicit time-stepping. Although well-conditioned, solving this linear system at each timestep easily dominates the rest of the computation in terms of cost. This makes the method unattractive for FWI.

To address the first issue, we point out that certain triangular finite element spaces do admit diagonal approximations to mass matrices. These spaces contain the standard set of polynomials of some degree $k$, enriched with certain bubble functions (Chin-Joe-Kong et al., 1999). For each such space, it is possible to identify a set of interpolation nodes that also can be combined with appropriate weights to define a sufficiently accurate quadrature rule. Thus, the Kronecker property of the basis functions

at the quadrature points leads to the quadrature rule delivering a diagonal mass matrix. SEM uses the same principle, using Gauss-Lobatto quadrature points as interpolation nodes on quadrilateral/hexahedrals meshes. Such sets of points are known up to $k = 5$ for triangles and $k = 4$ for tetrahedra and due to their diagonal mass matrix, they can be used for fast fully-explicit numerical wave simulations (Chin-Joe-Kong et al., 1999; Mulder et al., 2013a; Geevers et al., 2018b, a). However, to the authors' knowledge these elements have not been used in peer-reviewed literature to perform seismic inversions. Thus, several questions remain on how these elements may benefit the other components (e.g., discrete adjoint, sensitivity kernel calculation) of the seismic inversion posed in a finite element framework.

A second major difficulty is the generation and design of a variable resolution triangular mesh. This can be a potentially laborious mesh generation pre-processing step and can strongly limit the applicability of the method, especially in 3D (e.g., Anquez et al., 2019; Peter et al., 2011; Modave et al., 2015). To take full advantage of FEM, elements in the mesh must be sized in an optimal way to take into account numerical stability criteria, the numerical methods used, the seismic data (e.g., velocity model), and the characteristics of the forcing mechanism simultaneously. Further to this point, the most ubiquitous methods to triangulate the computational domain with simplices (e.g., Delaunay triangulation) suffers from the formation of degenerate elements termed slivers (Tournois et al., 2009), which would otherwise render a wave propagation simulation useless. Despite this, triangular mesh generation is generally preferred over hexahedral mesh generation as triangular meshes offer, in general, a greater degree of flexibility in resolving complex and irregularly-shaped geometry. In this work, we explore the effect of variable mesh resolution on the forward-state problem based on the source's peak frequency and seismic velocity medium (e.g., waveform adapted meshes) and use these mesh resolution guidelines to design meshes for FWI.

Third, the high complexity of implementing efficient unstructured FEM frequently discourages domain practitioners. Compared to finite difference methods, FEM require additional levels of coding complexity associated with mesh data structures, numerical integration, function spaces, matrix assembly, and sophisticated code optimizations for looping over unstructured mesh connectivity (Luporini et al., 2015, 2017). Re-implementing such tasks in a particular application context (e.g., FWI) do not constitute a major advancement. Recognizing this issue, many advanced software packages have been put forward, separating the concerns between low-level programming/implementation and the high-level mathematical formulation to more confidently write FEM codes for various application domains (Krischer et al., 2015; Modrak et al., 2018; Alnæs et al., 2015; Witte et al., 2019; Cockett et al., 2015; Rücker et al., 2017; Louboutin et al., 2019; Rathgeber et al., 2017). These approaches often present a programming environment in which data objects correspond to higher-level mathematical objects inherent to inverse problems and/or numerical discretizations such as the finite difference, finite element or finite volume methods. For example, packages have focused on creating high-level abstractions for geophysical inversion problems (e.g., Witte et al., 2019; Cockett et al., 2015; Rücker et al., 2017), while others more generally deal with solving variational problems using the finite element method (Rathgeber et al., 2017) or writing performant stencil codes for finite difference methods (Louboutin et al., 2019).

The Firedrake project (Rathgeber et al., 2017) is one example of a powerful programming environment that adequately address the code complexity inherent to FEM and leads to the development of computationally performant and highly technical FEM implementations in concise scripts within the Python programming language. Firedrake, like FEniCS (Alnæs et al.,

2015), uses the Unified Form Language (UFL Alnæs et al., 2014) to describe variational problems in mathematical syntax. This high-level symbolic description can be manipulated as a first-class object so that Jacobians and adjoint operators can be automatically derived (Alnæs et al., 2014; Farrell et al., 2013) and, as recently shown by Farrell et al. (2020), time discretization can be automated from a semi-discrete problem description. Although written in Python, Firedrake internally generates efficient low-level code and interfaces to advanced solver packages and hence can scale to billions of DoF (Kirby and Mitchell, 2018; Farrell et al., 2019). This combination of high-level features and performance makes Firedrake an interesting candidate for developing an extensible and maintainable code stack for performing FWI with finite element methods.

The aim of this paper is to address the issues associated with the application of triangular, unstructured FEM to perform FWI with the higher-order mass lumped elements of Chin-Joe-Kong et al. (1999) and Geevers et al. (2018b). We demonstrate the concept thatwaveform adapted meshes combined with a discrete adjoint technique lead to an FWI implementation that requires significantly fewer computational resources while maintaining the accuracy of the result. Several technical aspects of the methods are detailed including mesh-dependency, domain truncation, efficient mesh design, and gradient-based adjoints providing practical information for FWI implementations using finite element methods and making triangular finite element methods more attractive for future applications in seismic imaging applications. All developments detailed in this work are available in an open source Python implementation using the Firedrake programming environment named spyro (https://zenodo.org/record/5164113).

The article is organized as follows: first we introduce the FWI algorithm and discuss the continuous formulation. Thereafter, we focus on the discretization of the governing equations in both space and time. Following this, we discuss our Firedrake implementation. Then we study the error associated with discretizing the domain with variable resolution triangular meshes. Lastly, we demonstrate computational results in both 2D and 3D, discuss and conclude the work.

## 2 Full waveform inversion

Figure 1 shows a basic overview of an experimental configuration used in FWI in a marine environment. FWI is designed to simulate a geophysical survey and estimate the model parameters (e.g., seismic velocity) to explain the observed waveforms in a way that minimizes a measure of error (e.g., misfit). This process is known as inversion. In contrast to less computationally expensive tomography methods that use only the phase information of recorded signals, FWI utilizes both amplitudes and phase information from recorded data and can thus image higher resolution targets to half the spatial wavelength of the source frequency (Fichtner, 2011).

In a typical field setup in an offshore/marine environment, a ship tows a cable potentially several kilometers long with hundreds of microphones (Figure 1). Nearby the ship, small controlled explosions known as shots or sources are created. These shots propagate sound waves that interact with the subsurface medium and produce signals recorded by the microphones. The collection of seismic signals for a particular shot explosion event is referred to as a shot record and the quantity and the location of the sources with respect to the location of the receivers is referred to as acquisition geometry.

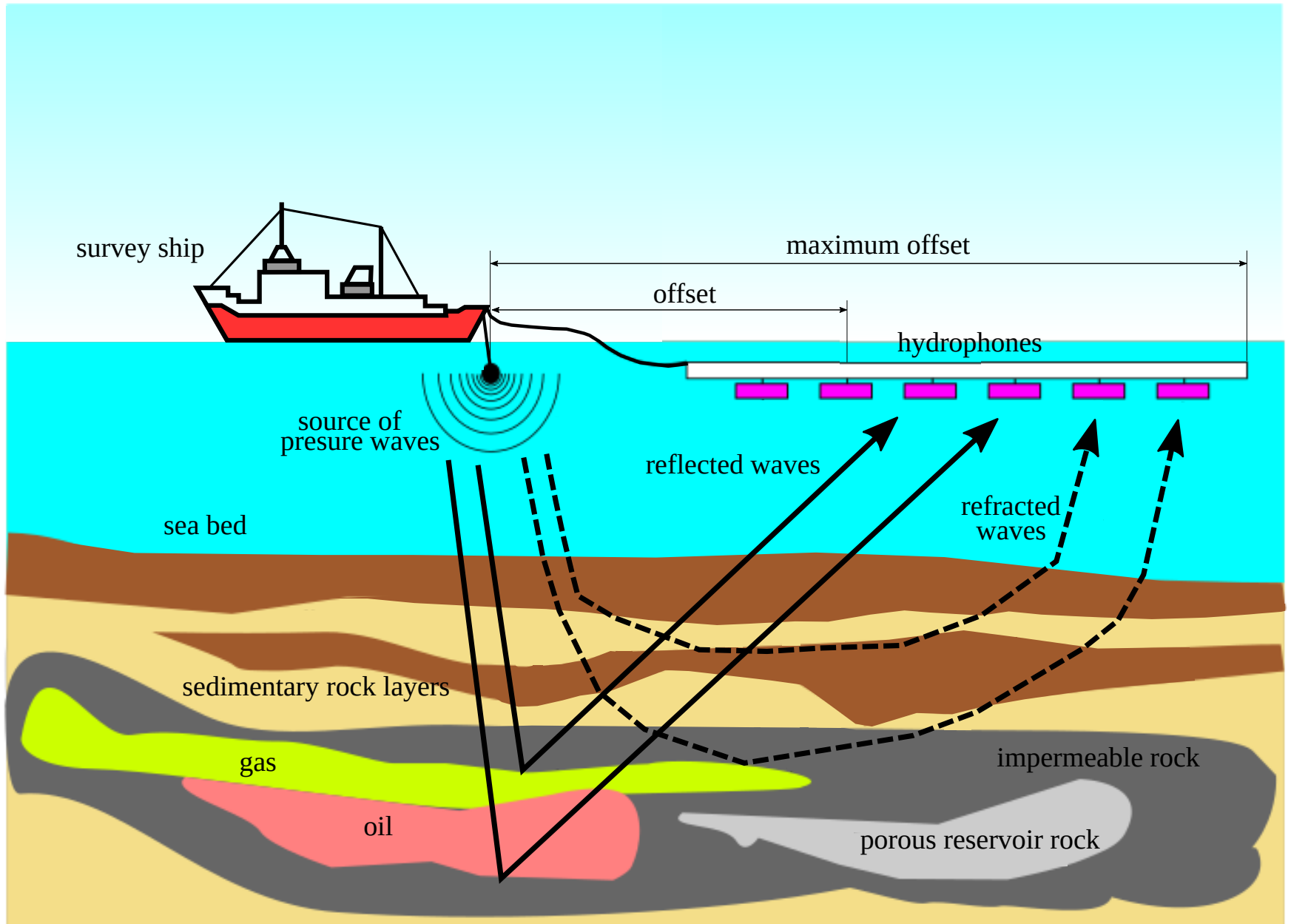

**Figure 1.** A simplified illustration of a marine seismic survey with relevant components annotated.

FWI can either be posed in the time domain or frequency domain (Virieux and Operto, 2009; Pratt and Worthington, 1990). In 2D, the frequency domain approach is regarded as the more computationally efficient approach (Brossier et al., 2009; Virieux and Operto, 2009). In 3D however, the computational effort and memory requirements associated with solving the system of equations in the frequency domain can become prohibitive and negatively affect parallel scaling efficiency. Thus, the time domain approach for FWI is still used in applications and remains technically relevant.

One key challenge associated with FWI and inverse problems in general is that they require a adequate starting velocity model to converge toward the global minimum of the misfit. In other words, the initial model should be able to predict the travel time of any arrival involved in the inversion to within half a period of the lowest inverted frequency when a classical least-squares misfit function based on the data difference is used otherwise the FWI will converge to a local minimum (e.g., Virieux and Operto, 2009). Typically these initial models are created through time travel tomography methods with manual inspection and edits (Lines and Newrick, 2004).

### 2.1 Forward wave simulation in a PML truncated medium

In this work, the acoustic wave equation in its second-order form is considered in either a 2D or 3D physical domain $\Omega_0$. The acoustic wave equation has one free parameter $c$ that is the spatially-variable compressional wavespeed otherwise referred to as the P-wavespeed. The acoustic wave equation is frequently used in FWI applications because its numerical solution is computationally inexpensive compared to the solution of the elastic wave equation while still yielding practically useful inversion results in some scenarios (Gras et al., 2019).

When simulated waves reach the extent of the domain, they create reflections generating signals that are deleterious for FWI applications since field data do not contain these signals. Thus, in this work an absorbing boundary layer referred to as a Perfectly Matched Layer (PML) is included as a small domain extension $\Omega_{\text{PML}}$ to attenuate the propagation of the outgoing waves and $\Omega \in \Omega_0 \cup \Omega_{\text{PML}}$. Note that the PML surrounds $\Omega_0$ on all but the water layer of the domain, shown in Figure 1. The domain is truncated with a non-reflective Neumann boundary condition in order to absorb some remaining oscillations there (Clayton and Engquist, 1977). All examples in this text rely on the usage of this acoustic wave equation in this configuration and further technical details about the PML formulation used can be found in Kaltenbacher et al. (2013).

The coupled system of equations for the modified acoustic wave equation with the PML are given by the residual operators $R_u, R_{\boldsymbol{p}}, R_\omega$ as:

$$R_u(u,\boldsymbol{p},\omega,f) \equiv \frac{\partial^2 u}{\partial t^2} + \operatorname{tr}\Psi_1 \frac{\partial u}{\partial t} + \operatorname{tr}\Psi_3 u + \det\Psi_1 \omega - \nabla\cdot(c^2\nabla u) - \nabla\cdot\boldsymbol{p} - f = 0, \quad (1)$$

$$R_{\boldsymbol{p}}(u,\boldsymbol{p},\omega) \equiv \frac{\partial \boldsymbol{p}}{\partial t} + \Psi_1\boldsymbol{p} + \Psi_2(c^2\nabla u) - \Psi_3(c^2\nabla\omega) = \boldsymbol{0}, \quad (2)$$

$$R_\omega(u,\boldsymbol{p},\omega) \equiv \frac{\partial \omega}{\partial t} - u = 0, \quad (3)$$

$$\partial_t u\big|_{t_0} = v\big|_{t_0} = 0, \quad (4)$$

$$\boldsymbol{p}\big|_{t_0} = \boldsymbol{0}, \quad (5)$$

$$\omega\big|_{t_0} = 0, \quad (6)$$

$$(\partial_t u + c\nabla u\cdot\boldsymbol{n})|_{\partial\Omega} = 0, \quad (7)$$

where $u(\boldsymbol{x},t) : (0,T)\times\Omega \to \mathbb{R}$ is the pressure at time t and position $\boldsymbol{x} = (x,y,z) \in \Omega$, $\omega(\boldsymbol{x},t) : (0,T)\times\Omega \to \mathbb{R}$ is an auxiliary scalar variable and $\boldsymbol{p}(\boldsymbol{x},t) = (p_x,p_y,p_z) : (0,T)\times\Omega \to \mathbb{R}^3$ is an auxiliary vector variable and $p_x$, $p_y$ and $p_z$ are the vector components, $c(\boldsymbol{x})$ is the P-wavespeed, $f(\boldsymbol{x},t)$ is the source term, and $\Psi_i$ and $\sigma_i$ are the damping matrices and functions, respectively. We remark that this formulation of the modified acoustic wave equation with the PML is the same as that originally designed by Grote and Sim (2010) and Kaltenbacher et al. (2013) and these formulations differ by what constitutes the spatially-varying velocity model, which either comes from the variation in density or the variation in bulk modulus.

In 2D, the modified acoustic wave formulation is simplified since $p_z$, $\omega$ and $\sigma_z$ vanish and it becomes:

$$R_u(u,\boldsymbol{p},f) \equiv \frac{\partial^2 u}{\partial t^2} + \operatorname{tr}\Psi_1 \frac{\partial u}{\partial t} + \operatorname{tr}\Psi_2 u - \nabla\cdot(c^2\nabla u) - \nabla\cdot\boldsymbol{p} - f = 0, \quad (8)$$

$$R_{\boldsymbol{p}}(u,\boldsymbol{p}) \equiv \frac{\partial \boldsymbol{p}}{\partial t} + \Psi_1\boldsymbol{p} + \Psi_2(c^2\nabla u) = \boldsymbol{0}, \quad (9)$$

where the boundary conditions remain unchanged. Only one vector-valued variable (e.g., $\boldsymbol{p}$) is additionally solved for each timestep. In both 2D and 3D for all experiments in this work, quadratic polynomial exponents are used to control the variations in the damping layer functions $\sigma_i$ which are used to form the damping matrices (e.g., $\Psi_1$, $\Psi_2$, $\Psi_3$) (Kaltenbacher et al., 2013). Note that $\sigma_i$ are zero inside the physical domain $\Omega_0$.
Open Access EGU

All sources $f$ are forced with a time-varying Ricker wavelet with a specified peak frequency in Hertz. More details regarding the implementation of the source are provided later in Section 4.2.

### 2.2 Continuous optimization problem formulation

In this section, the optimization components of the FWI process are detailed. Experimental data are generated by exciting a physical domain $\Omega_0$ by $N_s$ independent shots, which are located at points $\{\boldsymbol{x}_i^s\}_{i=1,\cdots,N_s}$. For each shot $\boldsymbol{x}_i$, data is collected at an array of $N_m$ measurement points (receivers c.f., Figure 1) $\{\boldsymbol{x}_j^m\}_{j=1,\cdots,N_m}$ for a time interval of length $T$; for instance, $u_i(\boldsymbol{x}_j^m,t)$ for $t \in [0,T)$. As mentioned earlier, the collection of this time series data at an array of receivers produces what is commonly referred to as a shot record. The cost functional that represents the error between a given numerical experiment and the reference data (denoted here by $\tilde{u}$) is given by:

$$J = \frac{1}{2}\sum_{i=1}^{N_s}\sum_{j=1}^{N_m}\int_0^T (u_i(\boldsymbol{x}_j,t) - \tilde{u}_i(\boldsymbol{x}_j,t))^2 dt = \frac{1}{2}\sum_{i=1}^{N_s}\sum_{j=1}^{N_m}\int_0^T\int_\Omega (u_i(\boldsymbol{x},t) - \tilde{u}_i(\boldsymbol{x},t))^2 \delta_{\boldsymbol{x}_j} d\boldsymbol{x}dt \tag{10}$$

where the last equality is obtained by using the following property of the Dirac masses $\delta_{\boldsymbol{x}_j}$, acting on the points $\boldsymbol{x}_j$ (c.f., Brezis (2011)):

$$\int_\Omega f(\boldsymbol{x})\delta_{\boldsymbol{x}_j} d\boldsymbol{x} = f(\boldsymbol{x}_j), \tag{11}$$

where $f$ is a function smooth enough for the pairing to make sense.

For a given velocity model $c$ upon integration of equations (1), (2), and (3) or (8) and (9), we can compute the cost functional $J$. The goal of FWI is to find a velocity model $c$ that minimizes $J$. This problem is a PDE-constrained optimization problem that will be solved using a gradient-descent method. The gradient of $J$ with respect to $c$ otherwise referred to as the sensitivity kernel or the gradient can be posed in the Lagrangian formalism. For that, the Lagrangian is defined as:

$$\mathcal{L}(\{u_i,\omega_i,\boldsymbol{p}_i\},\{u_i^\dagger,\omega_i^\dagger,\boldsymbol{p}_i^\dagger\},c) = J(u_i) + \sum_{i=1}^{N_s}\int_0^T u_i^\dagger R_u(u_i,\boldsymbol{p}_i,\omega_i,f_i) + \sum_{i=1}^{N_s}\int_0^T\int_\Omega \boldsymbol{p}_i^\dagger \cdot R_{\boldsymbol{p}}(u_i,\boldsymbol{p}_i,\omega_i) + \sum_{i=1}^{N_s}\int_0^T\int_\Omega \omega_i^\dagger R_\omega(u_i,\boldsymbol{p}_i,\omega_i). \tag{12}$$

This Lagrangian is dependent on the forward solution $\{u,\boldsymbol{p}_i,\omega_i\}$, on the velocity model $c$ (e.g., the control variable) and also on the adjoint solution $\{u^\dagger,\boldsymbol{p}_i^\dagger,\omega_i^\dagger\}$. The optimal condition is verified if the variation of the above Lagrangian with respect to the forward, adjoint and control variable are zero. The variation of the Lagrangian with respect to the adjoint field will lead to the equations (1)-(3). Setting the variation of the Lagrangian with respect to the forward field to zero will lead to the adjoint

equations:

$$R_u^\dagger(u,\boldsymbol{p},\omega,f) \equiv \frac{\partial^2 u^\dagger}{\partial t^2} - \operatorname{tr}\Psi_1 \frac{\partial u^\dagger}{\partial t} + \operatorname{tr}\Psi_3 u^\dagger - \omega^\dagger - \nabla\cdot(c^2\nabla u^\dagger) - \nabla\cdot(c^2\Psi_2\boldsymbol{p}^\dagger) + \sum_{j=1}^{N_m}(u(t)-\tilde{u}(t))\delta_{\boldsymbol{x}_j^m} = 0, \quad (13)$$

$$R_{\boldsymbol{p}}^\dagger(u,\boldsymbol{p},\omega,f) \equiv -\frac{\partial \boldsymbol{p}^\dagger}{\partial t} + \Psi_1\boldsymbol{p}^\dagger + \nabla u_i^\dagger = \boldsymbol{0}, \quad (14)$$

$$R_\omega^\dagger(u,\boldsymbol{p},\omega,f) \equiv -\frac{\partial \omega^\dagger}{\partial t} + \det\Psi_1 u^\dagger + \nabla\cdot(c^2\Psi_3\boldsymbol{p}^\dagger) = 0. \quad (15)$$

In 2D, these equations become:

$$R_u^\dagger(u,\boldsymbol{p},\omega,f) \equiv \frac{\partial^2 u^\dagger}{\partial t^2} - \operatorname{tr}\Psi_1 \frac{\partial u^\dagger}{\partial t} + \operatorname{tr}\Psi_2 u^\dagger - \nabla\cdot(c^2\nabla u^\dagger) - \nabla\cdot(c^2\Psi_2\boldsymbol{p}^\dagger) + \sum_{j=1}^{N_m}(u_i(t)-\tilde{u}_i(t))\delta_{\boldsymbol{x}_j^m} = 0, \quad (16)$$

$$R_{\boldsymbol{p}}^\dagger(u,\boldsymbol{p},\omega,f) \equiv -\frac{\partial \boldsymbol{p}^\dagger}{\partial t} + \Psi_1\boldsymbol{p}^\dagger + \nabla u_i^\dagger = \boldsymbol{0}. \quad (17)$$

In addition to these volume-equations, we can deduce the boundary and initial/final conditions for their variables. One can verify that a homogeneous final condition (on $t = T$) has to be imposed in all variables $u^\dagger$, $\boldsymbol{p}^\dagger$, $\omega^\dagger$. Also, since the forward solution needs to satisfy the boundary conditions $\boldsymbol{n}\cdot\nabla u = 0$ and $\boldsymbol{n}\cdot\boldsymbol{p} = 0$ (which also has to be verified for the test functions $\delta u, \delta\boldsymbol{p}$), the adjoint variables admits the boundary conditions, which are the same for 2D and 3D:

$$\boldsymbol{n}\cdot\nabla u^\dagger + \boldsymbol{n}\Psi_2\boldsymbol{p}^\dagger = c^{-1}\partial_t u^\dagger, \quad \boldsymbol{n}\cdot(\Psi_3\boldsymbol{p}^\dagger) = 0, \quad \boldsymbol{x}\in\partial\Omega. \quad (18)$$

So the variation of the Lagrangian with respect to the control variable $c$, while keeping all the other variables constant, leads to the sensitivity kernel (or the gradient) $dJ/dc$:

$$\lim_{\varepsilon\to 0}\frac{\mathcal{L}(c+\varepsilon\delta c)-\mathcal{L}(c)}{\varepsilon} = \frac{d\mathcal{L}}{dc}\delta c \equiv \int_\Omega \frac{dJ}{dc}\delta c\, d\boldsymbol{x} = \sum_{i=1}^{N_s}\int_0^T\int_\Omega 2c\nabla u_i^\dagger\cdot\nabla u_i\,\delta c\, d\boldsymbol{x}\, dt. \quad (19)$$

where the terms involving the PML are not present in the physical domain $\Omega_0$ since the damping functions $\sigma_i$ are zero outside of the PML where we perform the optimization. The calculation of the sensitivity kernel and cost functional can then be used in an optimization algorithm of choice.

## 3 Numerical discretization

### 3.1 Spatial discretization

We have discretized the modified acoustic equation (Eq.(1)-(3), Eq.(8)-(9)) and their respective discrete adjoints (Eqs.(13)-(15), (16)-(17)) with a continuous Galerkin (CG) FEM. While the physical features of the velocity model in reality are likely discontinuous, CG FEM can still provide good approximate solutions to velocity modeling building, which often commence from smooth initial material parameters.

CG methods actually provide a family of methods, parameterized over the choice of approximating spaces rather than a single method. Frequently, the choice of approximating spaces only affects the overall accuracy – by choosing standard $P^k$ elements based on polynomials of degree $k$, one obtains a certain order of convergence. However, special choices of these approximating spaces may affect other aspects of the method. In particular, by using the elements that we describe later on, we obtain a so-called lumped mass matrix on each simplex, which obviates the need to solve a linear system for each explicit timestep.

Regardless of the particulars, we denote the finite element function space used within our CG method as $V^C$, spanned by some locally constructed basis $\{\phi_i(\boldsymbol{x})\}$. This will be used to discretize the pressure $u$, together with each component of the auxiliary vector $\boldsymbol{p}_i$ and possibly the variable $\omega$ if a 3D domain is considered. If we let $U, P$ and $Y$ be the vectors containing the weights of the projection of $u, \boldsymbol{p}$ and $\omega$ onto the FEM space $V^C$, the space-discrete equations can be cast in the following general matrix form (here only the 3D equations are presented, but the 2D case is analogous):

$$\mathbb{M}_u \ddot{U}_i + \mathbb{M}_{u,1} \dot{U}_i + \mathbb{M}_{u,3} U_i + \mathbb{M}_{\omega,1} Y_i + \mathbb{K} U_i + \mathbb{D} P_i = \mathbb{M}_u F_i, \tag{20}$$

$$\mathbb{M}_p \dot{P} + \mathbb{M}_{p,1} P + \mathbb{D}_{u,2} U_i - \mathbb{D}_{\omega,3} Y_i = 0, \tag{21}$$

$$\mathbb{M}_\omega \dot{Y}_i - \mathbb{M}_\omega U_i = 0, \tag{22}$$

where the matrices $\mathbb{M}_u$, $\mathbb{M}_{u,1}$, $\mathbb{M}_{u,3}$, $\mathbb{M}_p$, $\mathbb{M}_{p,1}$, $\mathbb{M}_\omega$ and $\mathbb{M}_{\omega,1}$ are mass-like matrices that do not involve any spatial derivative. The matrix $\mathbb{D}$ is the discrete divergence operator and $\mathbb{D}_{u,2}$ and $\mathbb{D}_{\omega,3}$ are gradient-like discrete operators. The matrix $\mathbb{K}$ is the stiffness matrix. The precise mathematical definitions of the matrices are given in A.

### 3.2 Higher-order mass lumping

For linear triangular elements, mass lumping can be accomplished using the standard Lagrange basis functions and vertex-based Newton-Cotes integration rule. However for higher-degree ($k > 1$) triangular elements, a similar approach leads to unstable and/or inaccurate methods. Higher-order triangular elements and associated quadrature rules that do admit a lumping quadrature scheme are given in Geevers et al. (2018a); Chin-Joe-Kong et al. (1999); Geevers et al. (2018b). The function spaces for these elements do not consist solely of polynomials of degree $k$, but also include certain higher-order bubble functions. These higher-order bubble functions increase the total number of degrees-of-freedom per element relative to traditional $P_k$ elements, but in explicit time-stepping contexts, the gain of having a diagonal mass matrix more than offsets this cost (e.g., Geevers et al., 2018b; Mulder and Shamasundar, 2016).

The aforementioned concept of using higher-order bubble functions to achieve these elements is illustrated and compared with standard Lagrange elements in both 2D and 3D in Figure 2 and Figure 3, respectively. These elements are referred to here as mass lumped (ML) elements. For example, $ML1tri$ and $ML1tet$ denotes degree-1 triangular and tetrahedral elements where the "$tri$" or "$tet$" refers to a triangular or tetrahedral element, respectively.

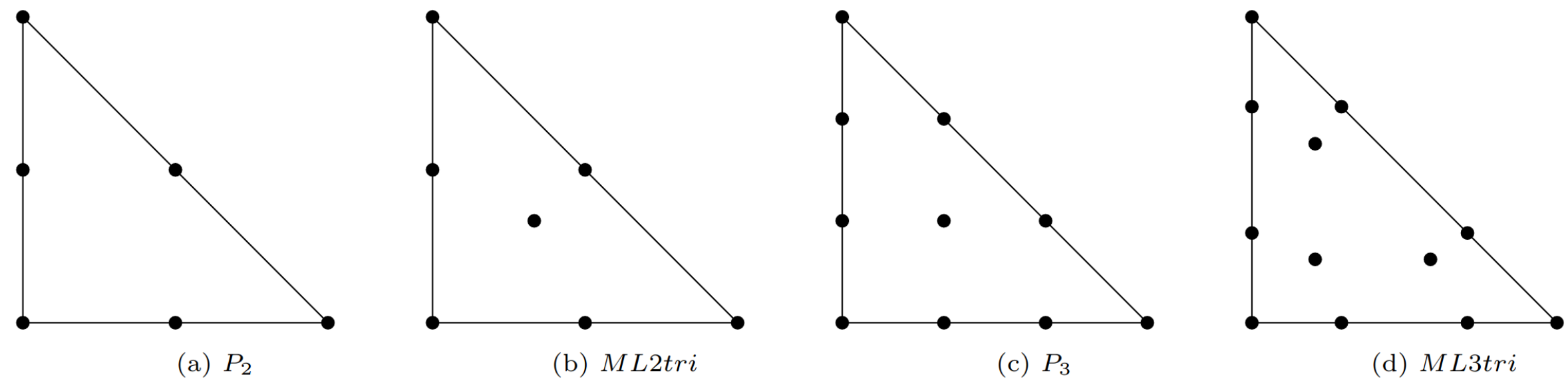

**Figure 2.** Some two-dimensional Lagrange and ML elements

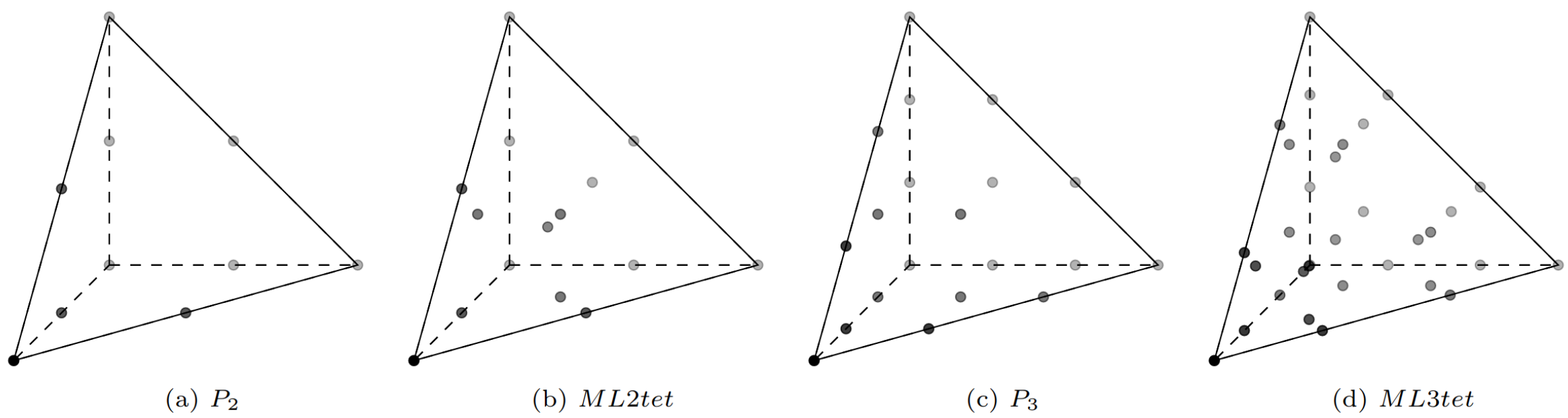

**Figure 3.** Some three-dimensional Lagrange and ML elements.

### 3.3 Waveform adapted triangular meshes

In order to efficiently discretize the domain, a triangular mesh of conforming elements interchangeably referred to as a mesh has to be generated. The major benefit of this approach is that mesh elements range in size according to several aspects elaborated below (e.g., Figure 4), reducing the total number of DoF. On the contrary, for structured grids the design of the elements is fully controlled using a regular structured mesh. While a structured grid greatly simplifies applications, they impose the additional computational cost of dramatically over resolving some areas of the domain from the standpoint of minimizing numerical error and dispersion.

The design of a so-called “optimal" mesh in a way that maximizes accuracy while minimizing computational cost through mesh size variation represents a challenging task. One crucial aspect is the numerical stability condition, which puts constraints on meshing because the timestep is affected by the smallest cell via the CFL condition (e.g., Mulder et al., 2013b). It is crucial therefore that the mesh generation program ensures elements are as large as possible to avoid prohibitively small simulation timesteps. Mesh size variation must also be gradual in order to minimize numerical error (Persson, 2006).

In this work, variable resolution element sizes are based on the acoustic wavelength, the CFL condition, and a mesh gradation rate. Altogether the design of resolution becomes proportional to the wavelength of the acoustic wave hence the phrase waveform adapted. The assumption is made that all triangles will be nearly equilateral, which is necessary for accurate sim-

Geoscientific Model Development Discussions Open Access EGU

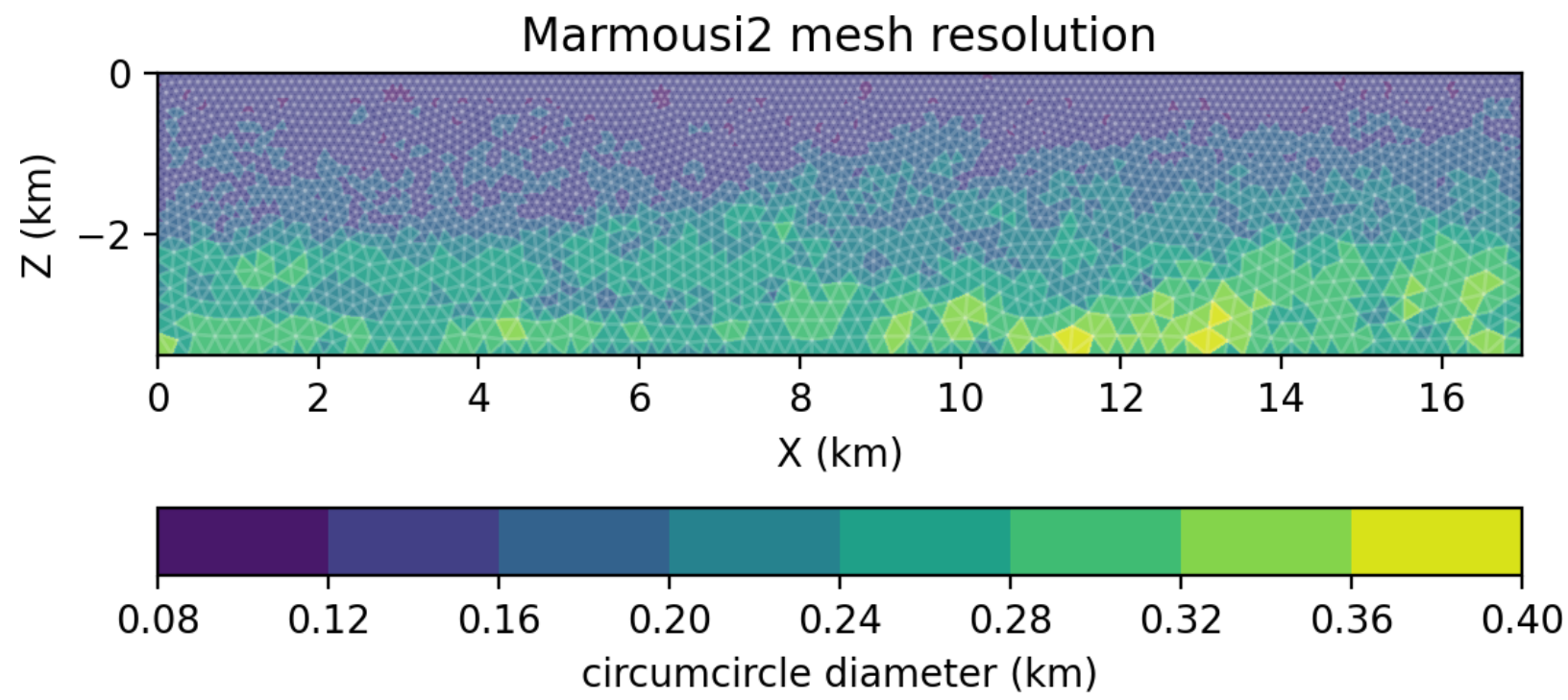

**Figure 4.** The Marmousi2 P-wavespeed model (Martin et al., 2005) discretized using a graded mesh with 4 cells per wavelength ($C = 4$) for a Ricker source with a peak frequency of 5 Hz. The mesh contains $3,022$ vertices and $5,743$ elements. The element size is the circumdiameter of each enclosed circle of each triangle.

ulation with FEMs. The mesh can be the result of any external mesh generator; in this work we use a domain specific mesh generator tool called SeismicMesh (Roberts et al., 2021) that is capable of generating 2D/3D triangular meshes with the vast majority of triangles that are approximately equilateral and with elements sized according to the local seismic velocity. The desired distribution of triangular edge lengths $l_e$ in our meshes are calculated using a ratio of the local seismic velocity (e.g., P-wavespeed) and the representative frequency of a source wavelet:

$$l_e(\boldsymbol{x}) \propto \frac{c(\boldsymbol{x})}{C \cdot f_{source}}, \tag{23}$$

where $c(x)$ is once again the spatially variable P-wavespeed, $f_{source}$ is the representative frequency of a source wavelet and $C$ denotes the number of cells-per-wavelength. An example of a typical mesh size distribution for the synthetic P-wavespeed model Marmousi2 (Martin et al., 2005) is shown in Figure 4. In the case of a marine domain such as Marmousi2, the layer of water along the top of the model must contain the finest mesh resolution since the acoustic wavelength is the shortest there. It is also important to point out that mesh sizes must be smoothly varying (otherwise referred to as a graded) to avoid numerical errors when simulations are performed. In this work, we use a mesh gradation rate of $15\%$, which was obtained through trial-and-error.

The length of the element's edges $l_e$ can be related to the cells-per-wavelength parameter $C = \lambda/l_e$, which in turn affects the number of grid-point-per-wavelength $G$ of a given problem. The parameters $C$ and $G$ are related to one another through:

$$G = \alpha(P) \cdot C \tag{24}$$

where $\alpha(P)$ is a constant coefficient that is a function of the spatial polynomial degree $k$. ML elements have a higher number of nodes-per-element, therefore they have a higher $\alpha$ per polynomial degree than standard Lagrange elements. Padovani et al. (1994) refers to $G$ as the average number of grid-points-per-space and not the maximum value of grid spacing inside the element between all possible pairwise nodal combinations. Therefore, $\alpha(P)$ is calculated based on the square root of the number of

DoF ($n_{DoF}$) per number of elements ($n_e$) in the mesh, $\sqrt{n_{DoF}/n_e}$. When this metric is applied to SEM quadrilateral elements, it gives results that match the values reported in Lyu et al. (2020).

The selection of $C$, and consequently $G$, raise several important questions such as: what is the minimal $G$ that can minimize numerical dispersion error and how does the choice of $G$ affect the total DoF for a given problem. These are important aspects as they yield significant effects on both the runtime and computational requirements of FWI with FEM and are later investigated in Section 5.2.

### 3.4 Time discretization

A second-order accurate fully-explicit central finite difference scheme was used to discretize all time derivative terms. While higher-order timestepping schemes such as a Dablain scheme (Dablain, 1986) or a Lax-Wendroff procedure (Lax and Wendroff, 1960) can be used, these methods were not pursued in this work to better focus the manuscript on the aforementioned issues concerning the spatial discretization and the usage of variable resolution triangular meshes. As a result, due to the usage of a relatively low-order timestepping scheme, we are required to select relatively small timesteps ($\Delta t \leq [1]ms$) ms to ensure the error from the time discretization is sufficiently small to study the effects on the spatial discretization on the forward-state wave propagation and FWI.

For a given variable $v_n = v(t_n)$ and for a given discrete time series $t_n = n\Delta t$, we have

$$\frac{dv}{dt}(t_n) \approx \frac{v^{n+1} - v^{n-1}}{2\Delta t}, \quad \frac{d^2v}{dt^2}(t_n) \approx \frac{v^{n+1} - 2v^n + v^{n-1}}{\Delta t^2}. \tag{25}$$

Using this discretization and also defining a state vector as a concatenation of all the variables $Q_i^n = [U_i^n, P_i^n, Y_i^n]^T$, the system of equations can be recast as

$$\mathcal{A}_{n+1}Q_{n+1} + \mathcal{A}_n Q_n + \mathcal{A}_{n-1}Q_{n-1} = \mathcal{M}F_n, \tag{26}$$

where those new matrices are given by:

$$\mathcal{A}_{n+1} = \begin{bmatrix} \Delta t^{-2}\mathbb{M}_u + (2\Delta t)^{-1}\mathbb{M}_{u,1} & 0 & 0 \\ 0 & (2\Delta t)^{-1}\mathbb{M}_p & 0 \\ 0 & 0 & (2\Delta t)^{-1}\mathbb{M}_\omega \end{bmatrix},$$

$$\mathcal{A}_n = \begin{bmatrix} -2\Delta^{-2}\mathbb{M}_u + \mathbb{M}_{u,2} + \mathbb{K} & \mathbb{D} & \mathbb{M}_\omega \\ \mathbb{D}_{u,2} & \mathbb{M}_{p,1} & -\mathbb{D}_{\omega,3} \\ -\mathbb{M}_\omega & 0 & (2\Delta t)^{-1}\mathbb{M}_\omega \end{bmatrix},$$

$$\mathcal{A}_{n-1} = \begin{bmatrix} \Delta t^{-2}\mathbb{M}_u - (2\Delta t^{-1})\mathbb{M}_{u,1} & 0 & 0 \\ 0 & 0 & -(2\Delta t)^{-1}\mathbb{M}_p \\ 0 & 0 & (2\Delta t)^{-1}\mathbb{M}_\omega \end{bmatrix}.$$

In order to solve for the variables at timestep $n+1$ given the previous ones, we need to invert $\mathcal{A}_{n+1}$, which is a mass-like matrix. While this requires significant work for standard $P_k$ elements, it is trivial for ML elements with the specialized quadrature rules discussed in Section 3.2.

In a practical application, it remains important to be able to determine a numerically stable timestep for this discretization and this depends on element degree $k$ and the quality of the mesh's elements. The maximum stable timestep can be estimated *a priori* by calculating the spectral radius of the scalar waves spatial operator while ignoring the contribution from the PML terms:

$$\mathbb{L} = \mathbb{M}_u^{-1}\mathbb{K} \tag{27}$$

A reasonable upper bound for the maximum stable timestep can then be found through (e.g., Mulder et al., 2013b):

$$\Delta t_{CFL} \leq \frac{2}{\sqrt{\rho(\mathbb{L})}} \tag{28}$$

where $\rho$ is the spectral radius estimated via Gershgorin's Disk Theorem (Geršgorin, 1931) and the subscript $CFL$ implies an upper bound on the timestep. This is possible to do explicitly for ML elements since $\mathbb{M}$ is diagonal and can be inverted onto $\mathbb{K}$ by just scaling rows.

In practice, a timestep $10\%$ to $20\%$ lower than the estimate provided by Eq. (28) remains stable and helps ensure numerical stability can be maintained throughout the inversion process as the seismic velocity is inverted. In 3D, the spectral radius $\rho(L)$ and consequently the maximum numerically-stable timestep are highly sensitive to the minimum dihedral angle in the mesh (Tournois et al., 2009). Thus, degenerate triangles termed slivers can result in exceedingly small numerically-stable timesteps and must be removed from the mesh. In practice, a minimum dihedral angle bound greater than $15°$ is often desired in order to maximize the stable timestep. However, this can be difficult to achieve in practice due to mesh generation challenges with variable resolution meshes. The minimum dihedral bound can be enforced in the mesh connectivity through a sliver-removal algorithm implemented in Roberts et al. (2021). All 3D meshes used in this work feature a minimum dihedral angle at least greater than $12°$.

### 3.5 The adjoint-state and gradient problems discretized

The numerical implementation of the adjoint problem and the gradient computation are detailed in this section. From the adjoint equations presented in their strong form (Eq.(13)-(15) and Eq.(16)-(17)), we then derive the associated variational formulation through the canonical procedure:

$$\int_{\Omega} \frac{\partial^2}{\partial t^2} u^\dagger v - \int_{\Omega} \frac{\partial}{\partial t} \operatorname{tr}\Psi_1 u^\dagger v + \int_{\Omega} \left(\operatorname{tr}\Psi_3 u^\dagger - \omega^\dagger\right) v + \int_{\Omega} c^2 \nabla u^\dagger \cdot \nabla v, \tag{29}$$

$$+ \int_{\Omega} c^2 (\Psi_2 \boldsymbol{p}^\dagger) \cdot \nabla v - \int_{\partial\Omega} c \partial_t u^\dagger v = - \sum_{j=1}^{N_m} (u(t, \boldsymbol{x}_j^m) - \tilde{u}(t, \boldsymbol{x}_j^m)) v(\boldsymbol{x}_j^m), \tag{30}$$

$$- \int_{\Omega} \frac{\partial}{\partial t} \boldsymbol{p}^\dagger \cdot \boldsymbol{q} + \int_{\Omega} (\Psi_1 \boldsymbol{p}^\dagger) \cdot \boldsymbol{q} + \int_{\Omega} \nabla u_i^\dagger \cdot \boldsymbol{q} = 0, \tag{31}$$

$$- \int_{\Omega} \frac{\partial}{\partial t} \omega^\dagger \gamma + \int_{\Omega} \det \Psi_1 u^\dagger \gamma - \int_{\Omega} c^2 \Psi_3 \boldsymbol{p}^\dagger \cdot \nabla \gamma = 0. \tag{32}$$

The discretization of this variational formulation can be cast as:

$$\mathbb{M}_u \ddot{U}_i^\dagger - \mathbb{M}_{u,1} \dot{U}_i^\dagger + \mathbb{M}_{u,3} U_i^\dagger - \mathbb{M}_\omega^T Y_i^\dagger + \mathbb{K} U_i^\dagger + \mathbb{D}_{u,2}^T P_i^\dagger = \mathbb{H}^T \mathbb{H} (U^n - \tilde{U}^n), \tag{33}$$

$$-\mathbb{M}_p \dot{P}^\dagger + \mathbb{M}_{p,1} P^\dagger + \mathbb{D}^T U_i^\dagger = 0, \tag{34}$$

$$-\mathbb{M}_\omega \dot{Y}_i^\dagger + \mathbb{M}_{\omega,1}^T U_i^\dagger - \mathbb{D}_{\omega,3}^T Y_i^\dagger = 0. \tag{35}$$

where $\mathbb{H}$ is the discrete version of the Dirac operator applied on all the measurement points in the domain. We remark that all the matrices used before (c.f., A) are reused but transposed (if not symmetric) both at the level of their entries and at the level of the equations. By discretizing the continuous equations with the FEM, we obtain the discrete adjoint. This is further clarified when discretizing Eq. (33) in time using the same procedure as before with central differences (e.g., Eq. (25)), which leads to the system for the variables in compact notation $Q^\dagger = (U^\dagger, P^\dagger, Y^\dagger)^T$:

$$\mathcal{A}_{n+1}^T Q_{n-1}^\dagger + \mathcal{A}_n^T Q_n^\dagger + \mathcal{A}_{n-1}^T Q_{n+1}^\dagger = \mathcal{H}^T \mathcal{H} (Q^n - \tilde{Q}^n).$$

In addition to the adjoint, the gradient is computed by discretizing Eq. (19) by letting the function $\delta c$ to be the trial function. The resulting linear system for the gradient, denoted $\mathcal{G}$ in its discrete form, is written as:

$$\mathbb{M} \mathcal{G} = \sum_{i=1}^{N_s} \sum_{n=1}^{N_t} \int_{\Omega} 2c \nabla u_i^\dagger(t_n) \cdot \nabla u_i(t_n) \, \delta c \, d\boldsymbol{x}. \tag{36}$$

In order to derive the discrete adjoint and gradient, the time integral appearing in the continuous formulation (i.e., in the cost functional $J$, and in the definition of the inner product, in the Lagrangian functional $\mathcal{L}$) were all replaced with discrete sums that did not consider a time integration. This is a similar approach as what was performed in Bunks et al. (1995). For this reason, no $\Delta t$ factor (or other time integration methods such as trapezoidal or Simpson's rule) are present.

Also, we stress here that, in order to obtain the gradient $\mathcal{G}$, we need to solve Eq. (36) by inverting a mass-matrix. This matrix comes from the fact that, in the continuous gradient derivation, Eq. (19), the inner product is chosen to be the classical $L^2$ inner product, which is represented by the mass-matrix in the discrete framework. This choice of inner product ensures that

the gradient will be mesh-independent (e.g., Schwedes et al., 2017) in the sense that local mesh refinements will not produce differences in the gradient (if the problem is sufficiently mesh-converged). For example, if the right-hand-side expression in Eq. (36) is readily used as the gradient, the mesh-dependency would be present as the space integration would only be present on the right-hand-side.

### 3.6 Gradient subsampling

Numerical simulations often require several thousand timesteps integrating over several seconds with the aforementioned numerical approaches to compute the discrete gradient. As a result, there are significant memory requirements for storing the forward-state solution that are necessary to calculate $\tilde{Q}$ and subsequently $\mathcal{G}$. By considering that the numerically stable timestep given by the CFL condition is generally much shorter than the Nyquist sampling frequency dictated by the maximum source frequency, our implementation allows for a subsampling approach to calculate $\mathcal{G}$ to reduce memory overhead. In other words, the forward-state can optionally be saved at every $r$ timesteps ($r \gg 1$), where $r$ is the subsampling ratio and subsequently the gradient is calculated every $r$ timesteps.

## 4 Computer implementation

In this section, important components of our implementation are explained. The Firedrake package is used to implement the numerical developments. The code (https://zenodo.org/record/5164113), datasets (https://zenodo.org/record/5172307) together with Firedrake (zenodo/Firedrake-20210810.0) were used for all experiments.

An additional layer of implementation is necessary for applications in seismic problems as there are operations to execute FWI that fall outside of the capabilities within the Firedrake package. At the current point of development, components of Firedrake that are essential to compute the gradient with automatic differentiation (AD) (e.g., Dolfin-adjoint (Mitusch et al., 2019)) were not ready to be used in our FWI code.

The Rapid Optimization Library (ROL; Cyr et al., 2017) is used to solve the inverse problem given a gradient, a cost functional, and a method to update the velocity model. The ROL library provides interfaces to and implementations of various algorithms for gradient-based, unconstrained and constrained optimization coupled with line-search conditions that satisfy the strong Wolfe conditions (Wolfe, 1969). This improves the robustness of our FWI code by using well-developed and tested algorithms. The C++ library ROL is called in our Firedrake Python codes via a Python wrapper code called pyROL (Wechsung and Richardson, 2019).

As mentioned earlier in this work, we exclusively rely on the second-order optimization method L-BFGS (Byrd et al., 1995), which includes information about the curvature of the misfit function in the optimization process (Eq. (10)). The benefit of using second-order optimization methods in FWI has been studied previously and shown to benefit the computational efficiency of FWI (e.g., Castellanos et al., 2014). Using pyROL and Firedrake, a conventional FWI approach can be written in several dozen lines of Python.

### 4.1 Implementation of higher-order mass lumped elements

Five triangular elements for spatial polynomial degrees $k \leq 5$ from Chin-Joe-Kong et al. (1999) and three tetrahedral elements for spatial polynomial orders up to $k = 3$ (Geevers et al., 2018b) were implemented inside Finite element Automator Tabulator package (FIAT Kirby, 2004). In particular, we use the latest documented $k = 3$ 3D tetrahedral element $ML3tet$ from Geevers et al. (2018b) with 50 nodes. This program FIAT is used by the Firedrake package to tabulate a wide variety of finite element bases.

The quadrature rules are key to defining the finite element basis, so we began by implementing these within FIAT. Then, to define the finite elements themselves, we must first construct the function space. This is done using two particular FIAT features described in Kirby et al. (2012). First, we use the `RestrictedElement` operation on a Lagrange element to remove bubbles on facets where enrichment occurs, and then use a `NodalEnrichedElement` to put in higher-order bubbles on those facets. Second, we must provide the dual basis, which is just a list of pointwise evaluation functionals associated with the ML quadrature points. In addition to these, like all other FIAT elements, we also provide a topological association of the degrees-of-freedom to facets, and this information is used at a higher abstraction level by Firedrake to build local-to-global mappings.

Certain standard boilerplate is required to expose a new FIAT element to the rest of Firedrake. First, the element, along with certain metadata, must be announced within the Unified Form Language. Then, it must also be wrapped into FInAT (Homolya et al., 2017), which is a layer that provides abstract syntax for basis evaluation and supports higher-order operations, such as tensor-products of elements or making vector-valued spaces such as used for our $\boldsymbol{p}$ variable. It is this layer, rather than FIAT itself, that interacts with Firedrake's form compiler, `tsfc` (Homolya et al., 2018). Within `tsfc`, we must also provide a binding between UFL names and FInAT classes. Hence, although we make changes to several packages, they are rather superficial beyond the FIAT implementation.

### 4.2 Receivers and sources

To probe the computational domain, functionality is required to both record the solution at a set of points (i.e., receivers) and to inject the domain with a time-varying wavelet (i.e., sources; Figure 1). Since the location of receivers does not necessarily match vertices exactly inside the mesh connectivity, the wave solution must be interpolated to the receiver locations. Interpolation of the wave solution to the receivers is carried out in the same space of the finite elements used to discretize the domain.

Source injection is the adjoint operator of interpolating the solution to the receivers. To execute both, a Dirac mass is integrated against the finite-element basis functions in the form of weights equal to the basis functions evaluated at the source (receiver) position inside the element that contains the source (receiver). This point force source is of the form $f = w(t)\delta_{\boldsymbol{x}}(x)$, where $w(t)$ denotes the wavelet and $\boldsymbol{x_s}$ the source or receiver position. For the 2D case, the contribution to $f_{g(j_s,k),l}$ is $\int_{\mathcal{T}_{j_s}} f\phi_{j_s,k}dx = w(t)\phi_{j_s,k}(x_s)$. Here $g(j_s,k)$ defines the local-to-global map from node $k$ in element $j$ containing the $s$ source/receiver to the global set of DoF. For the adjoint calculation, the source is forced at the receiver locations that recorded the solution in the forward-state problem.

### 4.3 The inversion process

We start with an initial distribution of P-wavespeed $c$ and solve the forward-state problem to obtain $Q$. With the misfit known, we then solve the adjoint-state problem and obtain $Q^\dagger$. With both $Q$ and $Q^\dagger$ known the discrete gradient $\mathcal{G}$ can be computed. Thus, the updated velocity model $c$ can be computed by:

$$c^{k+1} = c^k + \alpha^k s^k. \tag{37}$$

where $\alpha^k$ is the step length and $s^k$ is the search direction and the superscript $k$ denotes the iteration. The L-BFGS is used to compute search directions $s^k$ and ROL is used to calculate $\alpha^k$ (Byrd et al., 1995).

The discussed inversion process is shown in Algorithm 1.

**Algorithm 1** Inversion of P-wavespeed $c$.

$k \leftarrow 0$
Set maximum number of iterations $iter_{max}$
Set convergence tolerance $tol$
$c^0 \leftarrow$ initial velocity model
Compute cost functional $J$ ▷ Eq. (10).
**while** $(J > tol) \,||\, k < iter_{max.}$ **do**
    Solve the forward-state problem for $Q$ ▷ Eq. (26)
    Solve adjoint-state problem for $Q^\dagger$ ▷ Eq. (33).
    Evaluate discrete gradient $\mathcal{G}^k$ ▷ Eq. (36).
    Compute search direction $s^k$ ▷ L-BFGS via ROL.
    Choose step length $\alpha^k$ ▷ L-BFGS via ROL.
    Update model's velocity $c^k$ ▷ Eq. (37).
    $k = k + 1$

In order to ensure a sufficient decrease of the objective functional at each inversion iteration $k$, line-search conditions are employed that satisfy the strong Wolfe conditions (Wolfe, 1969). This line search is implemented inside the ROL library.

### 4.4 Wave propagators

The spatial and temporal discretizations detailed in Section 2.1 are programmed with Firedrake. Figure 5 illustrates the main functions: 'forward.py' and 'gradient.py' and how they work together. The forward wave propagator called 'forward.py' returns two quantities for a given source configuration: the $Q$ at the timesteps determined by the subsampling ratio $r$ (c.f., Section 3.5) and the the forward-state solution $\mathcal{H}Q$ at the receivers for all timesteps. The adjoint-state propagator takes as input the difference between measured and modeled data at the receivers locations (otherwise referred to as the misfit) and the forward-state solution $Q$. To conserve virtual memory, while the adjoint-state propagator executes, the function called 'gradient.py' discards $Q$ as the adjoint $Q^\dagger$ and subsequently $\mathcal{G}$ (Eq. (19)) are calculated reverse in time. Note that the adjoint wave propagator returns the gradient summed over the timesteps dictated by the subsampling ratio $r$ (c.f., Section 3.5).

We point out that all spatial discretizations are performed using matrix-free approaches, which are available in the Firedrake computing environment and this reduces run-time memory requirements (e.g, Homolya et al., 2017; Kirby and Mitchell, 2018).

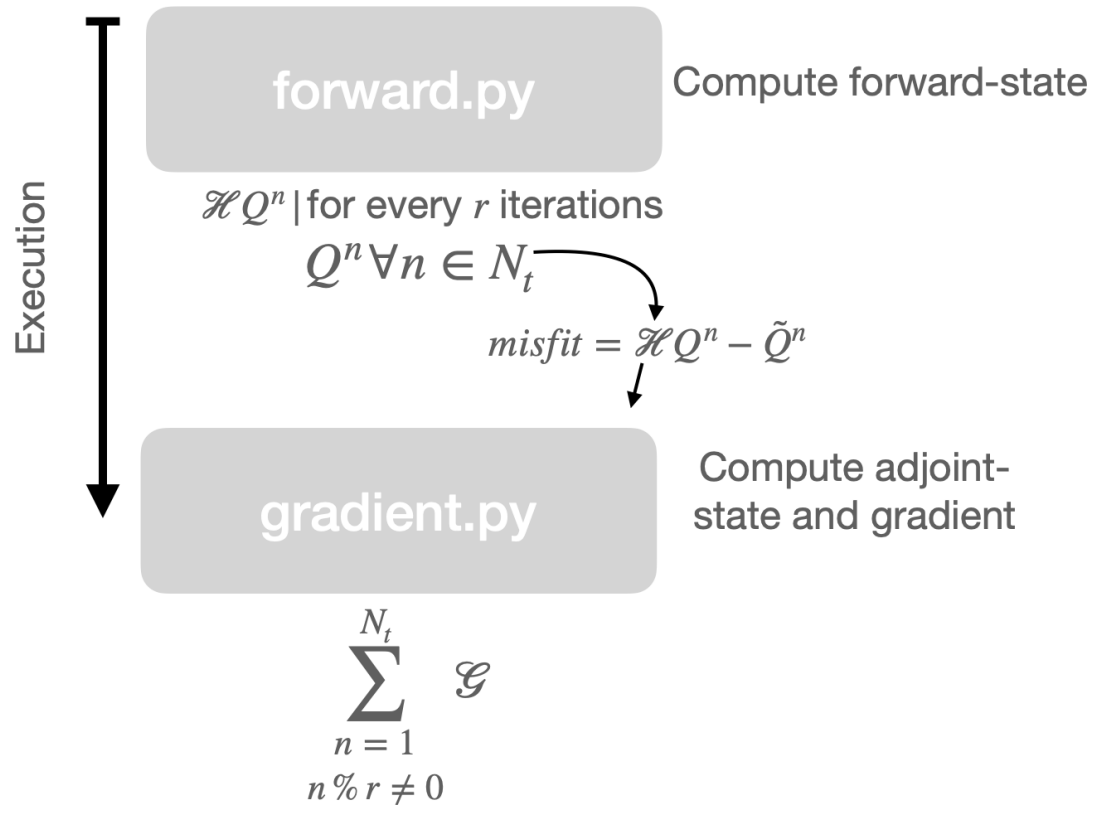

**Figure 5.** The functionality of the forward-state, adjoint-state and gradient Python codes.

### 4.5 Two-level parallelism strategy

A two-level parallelism strategy is implemented over both the sources and spatial domain decomposition. In space, domain decomposition parallelism is handled by the Firedrake library, which automatically handles setting up halo/ghost zones around each subdomain and performing the necessary communication at each timestep via the Message Passing Interface (MPI). In addition, Firedrake also provides options to configure the depth of the ghost layer for performance as needed. In this work, no ghost layers are added to the subdomains and instead the solution is shared only at the boundary nodes of each subdomain. At the source level, parallelism is trivial and handled by splitting the MPI communicator into groups of processes at initialization and assigning each group to simulate one source. Due to the usage of Firedrake, no additional code is required for parallelism as compared to the sequential version of the code.

### 4.6 Meshes and file I/O

Mesh files are read in from disk sequentially and then distributed in parallel if necessary; this functionality is handled latently by Firedrake. External seismic velocity models are read in from disk from a H5 file format at execution time. Gridded velocity data is bi-linearly interpolated onto the nodal DoFs of the elements of the mesh at runtime. In this way, seismic velocities can vary inside the element in the case higher-order elements are used. Gridded seismic velocity files can be prepared using Seimsicmesh (Roberts et al., 2021)

## 5 Computational Results

### 5.1 Numerical verification of discrete gradient

The accurate computation of the discrete gradient is crucial for the robustness of Algorithm 1. Through a numerical experiment, we demonstrate that the gradients computed through the optimize-then-discretize approach (c.f., Section 3.5) are approximately

Open Access EGU

| Case | $d^{co}$ | $d_h^{fd}$ | | |
|---|---|---|---|---|
| | | $h = 1e10^{-3}$ | $h = 1e10^{-4}$ | $h = 1e10^{-5}$ |
| 2D | 0.0681 | 0.0665 | 0.0664 | 0.0664 |
| 3D | 62.7069 | 65.4222 | 63.1321 | 62.9094 |

**Table 1.** Comparison of directional derivatives for 2D and 3D cases between the finite difference approximation ($fd$) and our discrete gradient ($co$) .

equal to the discrete gradients computed by the discrete gradients of the discrete objective functional. In this way, we compare the directional finite difference of the discrete objective functional. The finite difference directional derivative is given as a forward finite difference:

$$d_h^{fd}(c)(\tilde{c}) := \frac{J(c+h\tilde{c}) - J(c)}{h}, \tag{38}$$

where $\tilde{c}$ is the discrete direction for $c$ and $h$ is an arbitrarily small step size. The directional derivative obtained via the control problem is:

$$d^{co}(c)(\tilde{c}) = \tilde{c}^T \, \mathbb{M} \, \mathcal{G}. \tag{39}$$

Next, we verify that Eq. (38) and Eq. (39) produce accurate values for an arbitrary choice of $\tilde{c}$ considering the test problems displayed in Figure 6. For this test, the direction $\tilde{c}$ to test is that of the gradient $G$.

The considered 2D test problem to verify the numerical gradients was a physical domain $\Omega_0 = 1.0$ x $1.0$ km that features half the domain with a P-wavespeed of $4.0$ km/s and the other half with a P-wavespeed of $1.0$ km/s (Figure 6(a)). The physical domain was truncated with a 200 m PML on the sides while a non-reflective Neumann boundary condition was applied at the top. A $5$ Hz source is injected at $(-0.1, 0.50)$ km and the solution is recorded at $100$ receivers equispaced along a horizontal line at the bottom of the domain between $(-0.90, 0.1)$ km and $(-0.90, 0.90)$ km. In this configuration, the problem models a transmission of an acoustic wave. The domain was discretized with uniform-sized $ML2tri$ elements with $l_e = 20$ m in length yielding $G > 10$ given the $5$ Hz peak source frequency, which we found is sufficient for this experiment. The total duration of the simulation is $1.0$ seconds, which is long enough for the wave to be absorbed in the PML and transmitted to the receivers. A computational timestep $\Delta t = 0.50$ ms was used and the gradient was computed with all timesteps ($r = 1$). Note that the directional derivative (Eq. (39)) was integrated only in $\Omega_{physical}$ and masked in $\Omega_{PML}$.

In 3D a similar problem to the 2D case was considered within a physical domain $\Omega_0 = 1.0$ x $1.0$ x $1.0$ km. A 5 Hz source located was injected at $(0.1, 0.50, 0.50)$ km and the solution solution was recorded at a 2D grid of $100$ receivers equispaced apart in both $x$ and $y$ directions at the bottom end of the domain between $(0.90, 0.1, 0.1)$ km and $(0.90, 0.90, 0.90)$ km (Figure 6(b)). The domain was discretized with uniform $ML2tet$ with $l_e = 20$ m in length yielding $G > 10$. A computational timestep $\Delta t = 0.50$ ms is used and the gradient was computed with all timesteps ($r = 1$). Similar to the 2D case, the gradient was masked in $\Omega_{PML}$.

For both 2D and 3D cases, an initial velocity model with a uniform velocity of $4.0$ km/s was used; however, we simulated both the exact and initial models with the same mesh.

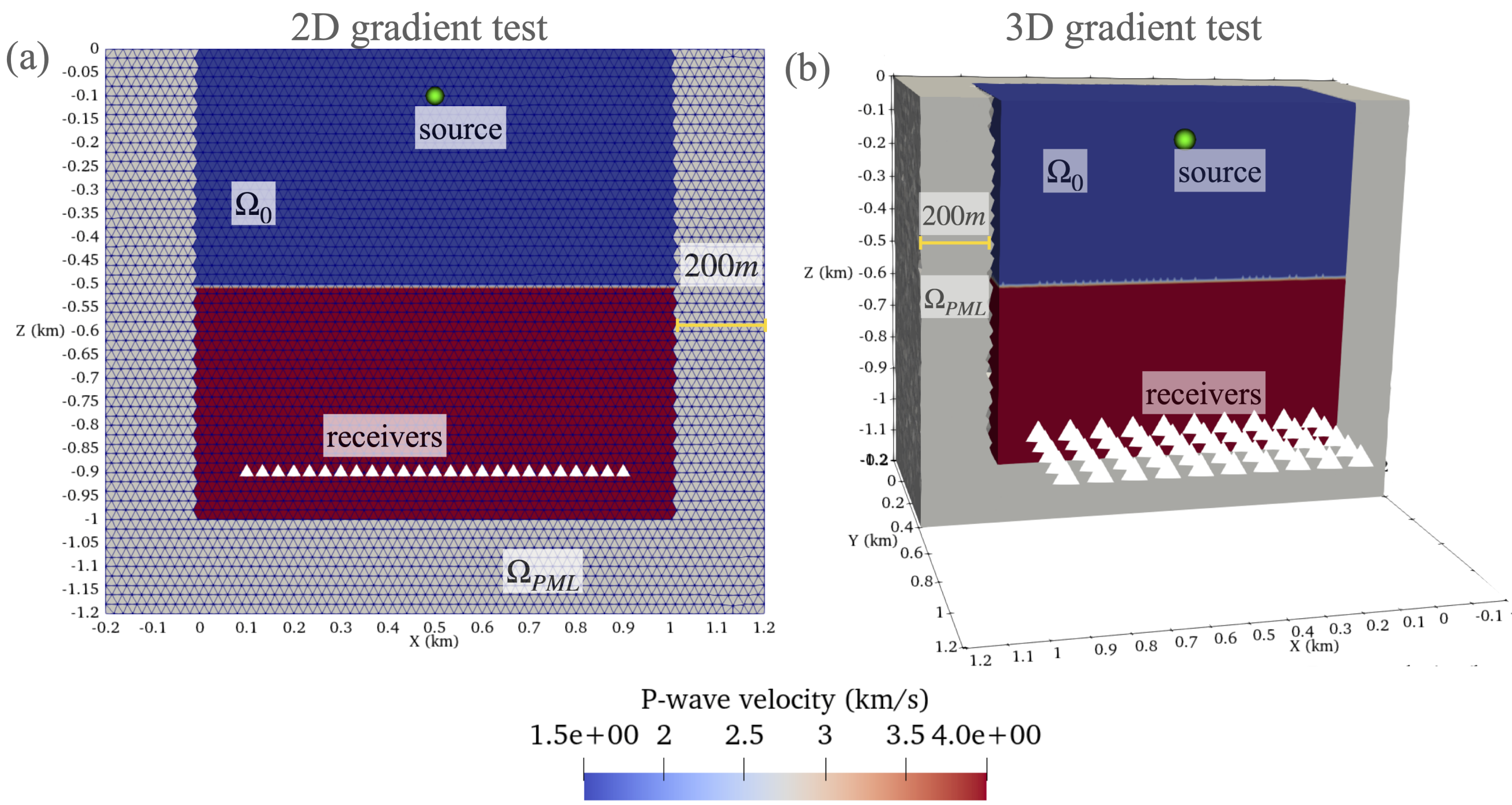

**Figure 6.** Problem configuration for verification of the discrete gradient in (a) 2D and (b) 3D. Note not all receiver positions are shown for visualization purposes.

We point out that relatively good agreement was found between the two derivatives in which the wavefield was resolved with at least $G = 10$ (Table 1), especially in the 2D case where the maximum relative difference between the gradients was less than $0.03$ % . In the 3D case, the discretization error becomes somewhat larger and results in maximum relative differences of, approximately $4.0$%, decreasing with smaller $h$.

## 5.2 On the design of waveform adapted meshes

To effectively apply higher-order mass lumped methods with unstructured meshes, it is important to understand the required mesh resolution for a given desired accuracy (Lyu et al., 2020; Geevers et al., 2018c). As mentioned in Section 4.1, ML elements contain a greater number of DoF-per-element than standard CG Lagrange or spectral/hp collapsed triangular elements. This does not imply, however, that a given problem would contain a greater number of DoF when discretized with ML elements since mesh resolution requirements for each method and element type vary widely (Lyu et al., 2020). Similar to the works of Geevers et al. (2018c) and Lyu et al. (2020), we investigate the accuracy of ML elements for forward-state wave propagation to guide their application in FWI.

#### 5.2.1 Reference wavefield solution

The implementation of the forward-state wave propagator in 2D and 3D was first verified in order to reliably intercompare solutions between elements. An equivalence was demonstrated between a "converged" numerical result computed on a highly-refined mesh in 2D and 3D and compared with their analytical solutions, respectively. Following that, the assumption was made that equivalence holds for all our subsequent tests implying that all the reference waveforms are "converged" numerical solutions, given sufficiently fine mesh resolution and sufficiently small numerical timesteps.

The method of manufactured solutions (MMS) was used to verify the implementation in accordance with a manufactured analytical solution. The manufactured 2D analytical solution was chosen as $t^2 sin(x) sin(y)$. In 3D the analytical solution was $t^2 sin(x) sin(y) sin(z)$. Both analytical solutions are defined on a unit square and unit cube with a 250 m wide PML layer. Numerical solutions were calculated on highly-refined reference meshes built with $G = 14.07$ using $ML5tri$ in 2D and $G = 9.30$ using $ML3tet$ in 3D. The velocity model was homogeneous with [1.43]km/s. The simulations used a timestep of $\Delta t = 1$ ms and were integrated for $0.10$ s. The MMS error was represented as the L2-norm between the analytical and numerical solution normalized by the analytical solution and only measured in the physical domain.

Our experiments demonstrated good agreement between the analytical and modeled solutions with relative error for the 2D reference homogeneous case of $0.34\%$ and for the 3D homogeneous case the relative error was $0.90\%$. These error values indicate the reference solutions represent numerically converged results given the spatial discretization and the forward-state code implementation is producing correct solutions.

#### 5.2.2 Homogeneous 2D P-wavespeed model

A 2D wave propagation experiment in a domain with a homogeneous velocity field was configured to quantify the accuracy of the forward-state solution with ML elements (Figure 7). The experiment is similar in design to that analyzed in Lyu et al. (2020), which used SEM of variable space order. A domain $40.0\lambda$ by $30.0\lambda$ (i.e., [11.4]km by [8.57]km) was generated where $\lambda$ is the wavelength of the acoustic wave given the model's wavespeed. The model had a uniform wave P-wavespeed of $1.43$ km/s, which is approximately the speed of sound in water. A Ricker wavelet with a peak source frequency of $5.0$ Hz was injected at the center of the domain and a grid of $36$ receivers was placed at a $10.0\lambda$ (i.e., $2.86$ km offset) to the right of the the source location in order to record and intercompare solutions (Figure 7). A $0.28$ km PML layer was added to absorb outgoing waves. The timestep used for each simulation was $20.0\%$ less than the $\Delta t_{CFL}$ estimated maximum stable timestep (c.f., Section 3.4). Meshes were generated for each element type by varying the $C$, which resulted in $G$ that ranged from $G = 2$ to $G = 10.5$. Results were compared against the solutions computed on the so-called reference meshes.

Error was calculated based on the simulated pressure recorded at the receivers locations with:

$$E = \sqrt{\frac{\sum_{r=1}^{N_r} \int_0^{t_f} (p_r - p_{r_{ref}})^2 dt}{\sum_{r=1}^{N_r} \int_0^{t_f} p_{r_{ref}}^2 dt}} \times 100\%, \tag{40}$$

where $N_r$ denotes the number of receivers, $t_f$ is the final simulation time in seconds, $p_r$ is the pressure at the receivers for a given mesh and $p_{ref}$ denotes the pressure at the receivers computed with a reference mesh. The time integration in Eq. (40)

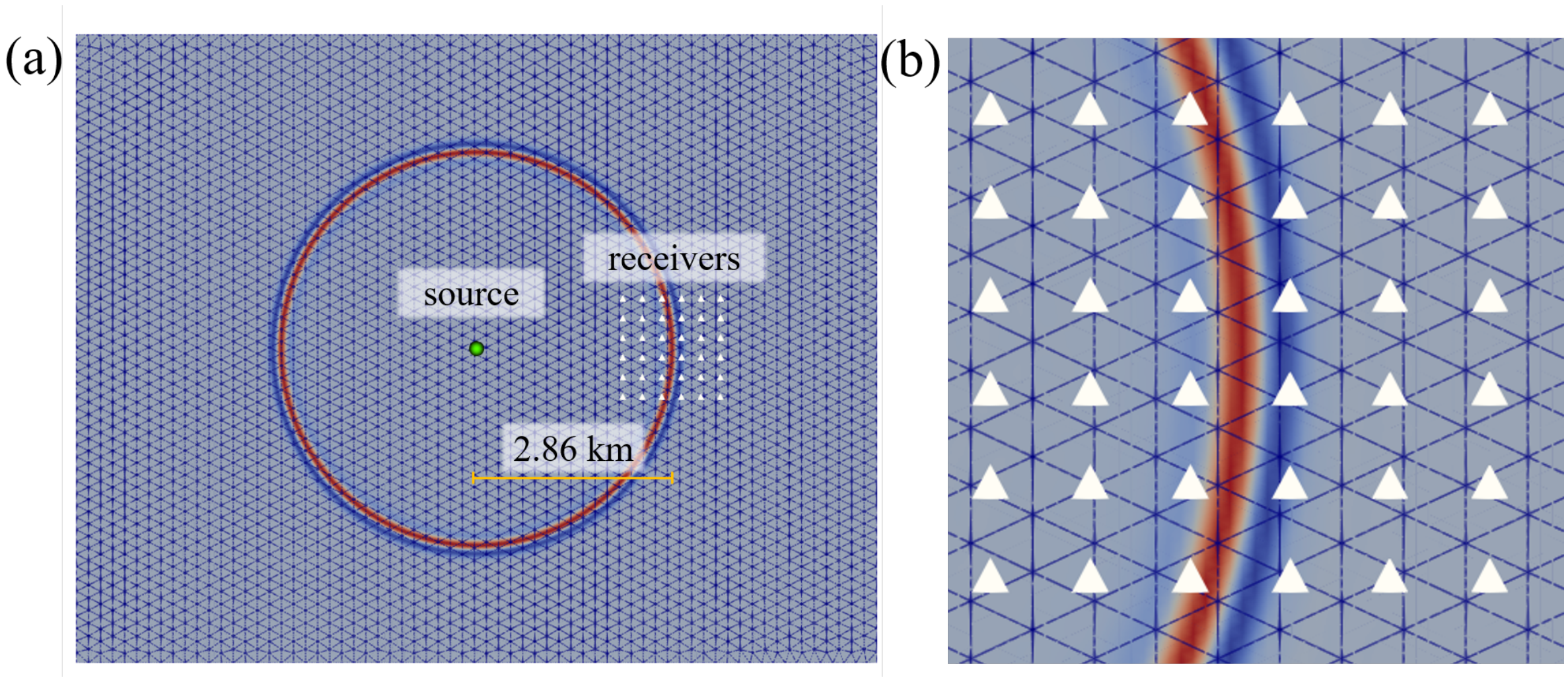

**Figure 7.** The experimental configuration to calculate the grid-point-per-wavelength $G$ values. In (a) the source is shown as a green circle and the receivers are denoted as white triangles with a close-up of the bin of receivers shown in (b). In both panels, the normalized wavefield is colored at $t = 2.25$ s.

was computed using the trapezoidal rule. Eq. (40) is a measure in percent difference between two solutions at the same set of receivers. It is important to point out that measuring $E$ in this way combines error associated with receiver and source interpolation as well as from the wave propagation. When $E$ is measured at one receiver at a particular offset coordinate $\boldsymbol{x}$, this is referenced by a subscript (e.g., $E_{\boldsymbol{x}=(0.5,0.5)}$); otherwise the quantity $E$ considers all receivers.

The space of $E$ for several values of $C$ and subsequently $G$ was explored for different ML elements in a process referred to as a grid sweep (Table 2). The objective of the grid sweep is to find the smallest $G$ (i.e., lowest grid point density) that can produce $E$ at or below a specified threshold. An allowable tolerance of $E = 5.0\%$ for each ML element was selected. While the $E = 5.0\%$ threshold chosen is arbitrary, it represents a measurement that can be used to intercompare solutions and, as we later show through application, to be sufficiently accurate for robust FWI. Further, smaller target thresholds for $E$ led to non-convergence for some elements. To execute the grid sweep, the value of $G$ was varied within a range of values depending on the change in $E$ in a similar manner to a back-tracking line search.

Overall, the homogeneous grid sweep results demonstrate that elements with spatial order $k > 2$ required fewer $G$ to achieve the same $E$ than $ML2tri$. As expected, the necessary $C$ in order to maintain the target $E$ decreases as spatial polynomial order is increased (Figure 8(b)). The relationship between $C$ and $G$ is not linear due to the higher-order bubble functions inside the ML elements (c.f., Section 3.2). Thus, the convergence rate of $E$ with respect to $G$ is not as consistent as it is for $C$ (Figure 8(a)).

Applying the results from the homogeneous grid sweep, the values for $G$ and $C$ that achieved $E = 5.0\%$ are shown in Table 2. The variation in $C$ that could achieve the $E$ threshold was $C = 1.69$ to $C = 5.85$ for $ML5tri$ to $ML2tri$, respectively, while for $G$ it varied between $G = 7.36$ and $G = 10.1$. The ML element that led to the smallest problem (e.g., minimum $G$) while

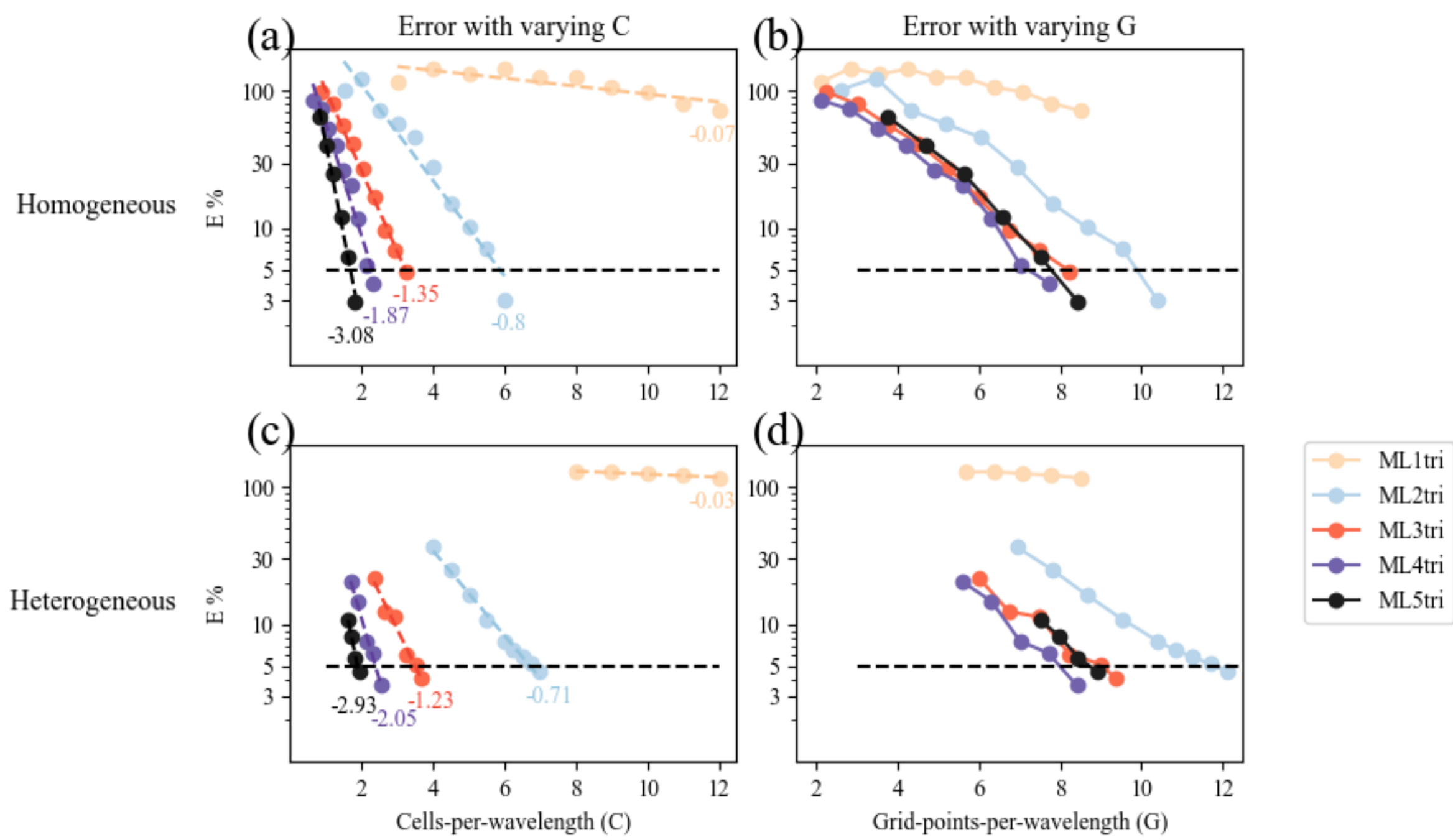

**Figure 8.** Panels (a) and (b) depict results for the homogeneous velocity model experiment to find the minimum $G$ (Section 5.2.2, Section 5.2.3). Panel (a) shows $C$ as a function of $E$ (Eq. 40). Panel (b) illustrates the $E$ as a function of $G$. Panels (c) and (d) show the same thing as (a) and (b) but for the heterogeneous velocity model experiment. Colored lines represent the spatial polynomial order of the element. The $E = 5\%$ threshold is drawn as a horizontal dashed black line on all panels. On panel (a), the line of best fit is drawn in the corresponding color and the slope of this line is annotated.

| | Homogeneous | | Heterogeneous | | $\Delta C$ |
|---|---|---|---|---|---|
| Element | minimum $G$ | minimum $C$ | minimum $G$ | minimum $C$ | % |
| $ML1tri$ | DNF | DNF | DNF | DNF | DNF |
| $ML2tri$ | 10.1 | 5.85 | 11.6 | 6.70 | 14.9% |
| $ML3tri$ | 7.86 | 3.08 | 9.06 | 3.55 | 15.3% |
| $ML4tri$ | 7.36 | 2.22 | 7.99 | 2.41 | 8.56% |
| $ML5tri$ | 7.88 | 1.69 | 8.54 | 1.84 | 8.38% |

**Table 2.** Results from the grid sweep for both the homogeneous and heterogeneous experiment to identify efficient values for $G$ using ML elements of varying spatial degree $k$ that maintain an error threshold of $E = 5\%$, as compared to a highly-refined reference solution (c.f., Section 5.2). Note that DNF stands for did not finish.

satisfying the target $E$ was $ML4tri$ with a $G = 7.36$, whereas $ML2tri$ required $G = 10.01$. It is important to note that the lowest order $ML1tri$ element performed poorly and did not achieve the target for $E$ with any configuration of $C$ tested.

#### 5.2.3 Heterogeneous 2D P-wavespeed model

In addition to generating a mesh that meets the requirements of the technique used to numerically discretize the PDE, the mesh must also account for local variations in the seismic velocity, which can have significant effects on the simulation of acoustic waves. In the case of simulation with a heterogeneous velocity model, $E$ combines errors associated with how the mesh discretely represents the local variations in velocity and errors associated with numerical discretization techniques. Thus, it is often necessary to add additional DoFs into the design of the unstructured mesh above what would be required for a homogeneous seismic velocity model to accurately represent local seismic features (e.g., Anquez et al., 2019; Seriani and Priolo, 1994; Lyu et al., 2020). However, it is important to point out that in FWI applications, the inversion commences from a smooth, initial velocity model (e.g, Fathi et al., 2015; Thrastarson et al., 2020; Trinh et al., 2019), with locations of velocity interfaces that are not generally not known prior.

As a result, in this experiment we added an additional percent to the parameter values of $C$ obtained from the homogeneous test case (Section 5.2.2). The percent difference in $C$ between homogeneous and heterogeneous results is defined as $\Delta C$.

$$\Delta C = \frac{(C_{het} - C_{hom})}{C_{hom}}, \tag{41}$$

where the subscripts $het$ and $hom$ denote the heterogeneous and homogeneous grid sweep results, respectively.

For triangular meshes, the $\Delta C$ that is necessary to minimize $E$ when simulating with heterogeneous velocity models has not been investigated in prior scientific literature to the authors' knowledge. It is also important to determine how the previously described homogeneous results can be applied to a heterogeneous seismic velocity model.

In a similar manner to the experiment with the homogeneous velocity model, a 2D experiment with a heterogeneous velocity model was performed for the BP2004 P-wavespeed model (Billette and Brandsberg-Dahl, 2005) (Figure 9). The BP2004 model represents geologic features in the Eastern/Central Gulf of Mexico and offshore Angola and is characterized by several salt bodies with P-wavespeeds $> 4$ km/s. The domain is 12.0 km by [67.0]km with an additional 1.00 km PML. A Ricker source was injected at $(-1.0, 34.5)$ km and a horizontal line of 500 receivers from $(-1.0, 36.5)$ km to $(-1.0, 44.5)$ km was used to record the wavefield solution. The acquisition geometry led to a near-offset of 2.00 km and a far-offset of 10.0 km from the source location, which are common dimensions in marine FWI applications for seismic velocity building (e.g., Virieux and Operto, 2009). Each simulation lasted 9.0 simulation seconds, which was sufficient time for reflected waves to reach the receivers with the largest offsets.

In the mesh generation process, a mesh gradation rate of 15.0% was enforced to bound the element size transitions (Figure 9). As with the homogeneous experiment, $G = 6$ to $G = 12$ were evaluated by comparing to the reference case. The reference case used a highly-refined mesh constructed with $G = 15.0$ and simulated with $ML5tri$ elements, which could correctly resolve all interfaces (Figure 9).

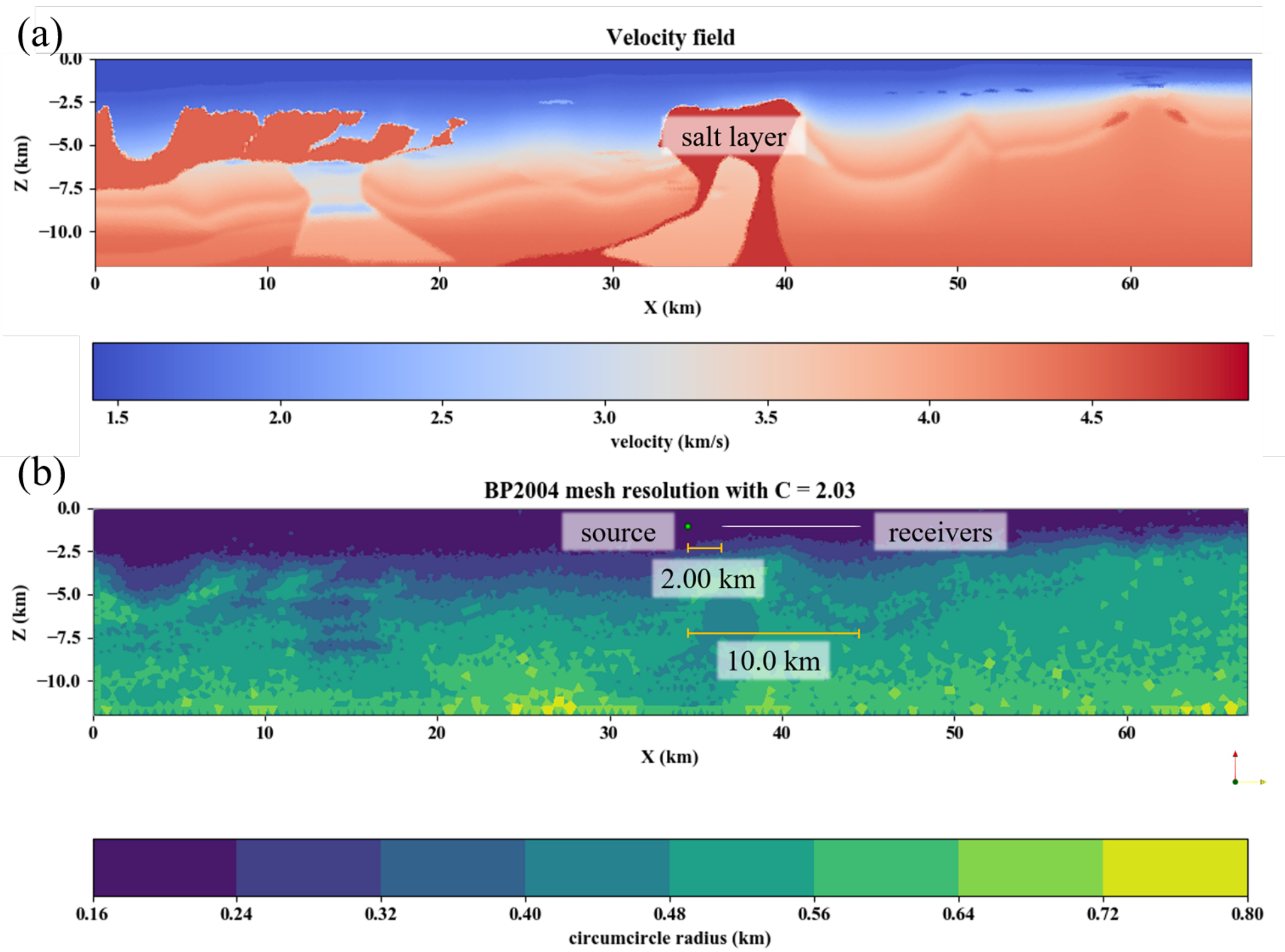

**Figure 9.** The reference problem configuration for the BP2004 seismic velocity model. Panel (a) shows the P-wavespeed data. Panel (b) shows the mesh resolution (circumcircle diameter) based on local adaptation of the mesh resolution to the P-wave data from the BP2004 velocity model shown in panel (a). The parameters used for mesh generation were $C = 2.03$, $ML5tri$, a velocity gradation rate of $15.0\%$, and a anticipated timestep of $\Delta t = 0.001$ ms.

As shown by Figure 8(c)-(d), the experiments with the BP2004 model consistently exhibited greater $E$ and slower convergence rates as compared to the values from the homogeneous experiment given the same $G$ (c.f., Figure 8(a)). As a result, the values for $C$ used to generate the meshes were increased from what was found in the homogeneous experiment by $\Delta C = 20.0\%$ and resulted in acceptable errors of $E = 3.45\%$, $E = 3.82\%$, $E = 3.44\%$, and $E = 3.38\%$, for $ML2tri$, $ML3tri$, $ML4tri$, and $ML5tri$ elements, respectively. $\Delta C$ less than $20.0\%$ did not sufficiently reduce the error to under the $E = 5.0\%$ threshold.

Wave propagation errors can be the result of dispersion and also by how well the mesh represents the local seismic wavespeed variations. In our mesh design, exact fault locations were not resolved with edge-orientated elements (e.g., Anquez et al., 2019) and our numerical discretization used elements from a continuous function space, thus error associated with the propagation of the reflected wavelet in the sharp contrast of the salt layer is expected. This error becomes more pronounced when using

larger element sizes associated with the higher-order ($k > 2$) ML elements. As an example of this, in Figure 10 $E$ is calculated individually for each receiver as a function of offset for $ML5tri$. A peak of $E = 7.71\%$ occured at the offset of 2.21 km that is associated with the reflection brought on by the salt layer Figure 11 and results in the peak $E$ not only in $ML5tri$ but also in $ML3tri$ and $ML4tri$. Neglecting the $E$ associated with the salt body reflection in this receiver location would reduce the error from $E = 7.71\%$ to $E = 2.02\%$. Furthermore, even though $E$ was kept below the previously defined threshold, a small dispersion error still exists and can be noted in receivers at the far-offset in all cases. Dispersion error was the most prevalent error only in the lower-order $ML2tri$ element, whereas in $ML3tri$, $ML4tri$, and $ML5tri$ the greatest error came from the wavelet reflected of the salt layer.

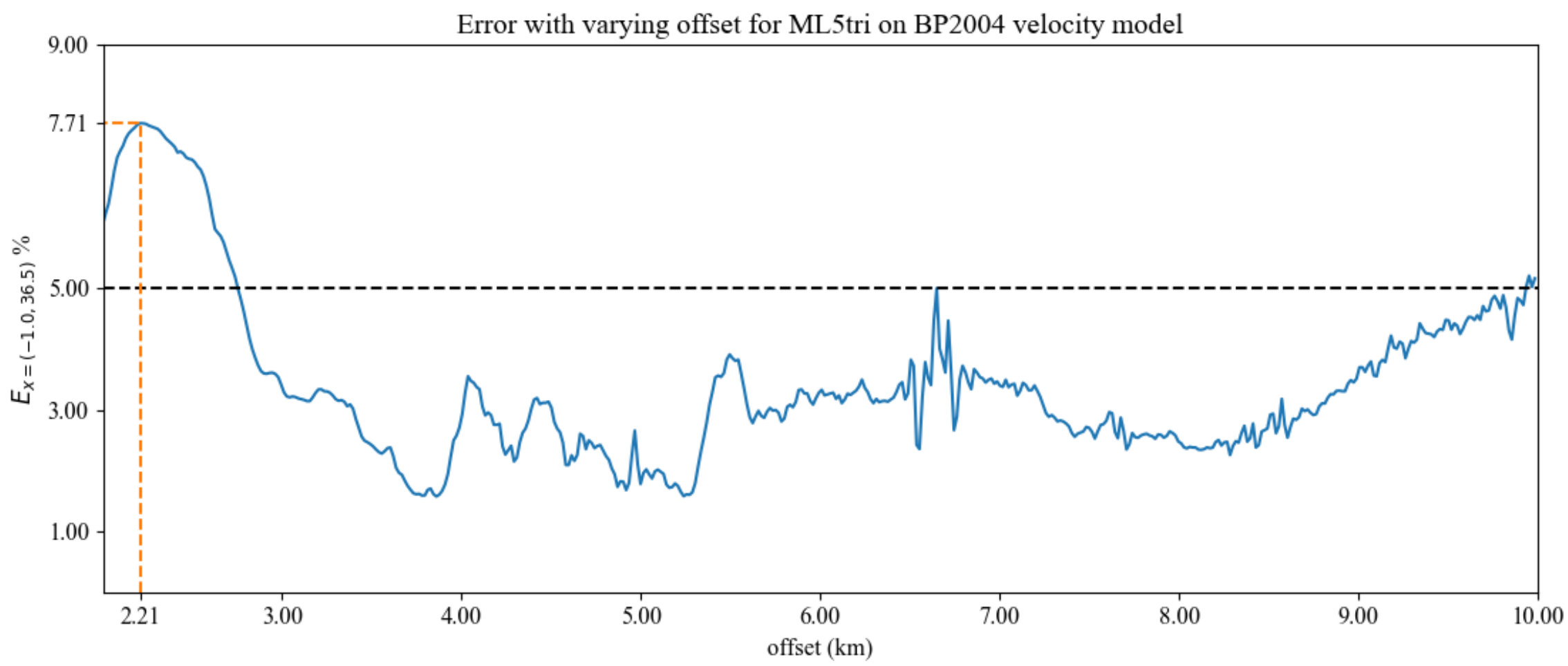

**Figure 10.** $E$ (Eq. (40)) as a function of the offset distance for $ML5tri$ in the heterogeneous model setup. Peak $E$ is annotated with dashed orange lines.

For $ML3tri$, $ML4tri$, and $ML5tri$ elements, peak $E$ stemmed from the reflected wave associated with the salt body due to the enlargement of element sizes nearby the salt body (Figure 11). Figure 11(b) illustrates the moment when the wave reflects off of the salt body and this reflected wave accounted for 73.8% of the total error at this receiver.

#### 5.2.4 Homogeneous 3D P-wavespeed

A similar experiment to that described Section 5.2.2 was used to assess 3D ML elements. The focus was placed on finding suitable values for $C$ and $G$ that minimize error for the $ML2Tet$ and the $ML3Tet$ elements that were discovered in Geevers et al. (2018c). Therefore, a homogeneous 3D model was created with uniform P-wavespeed of 1.43 km/s in a $15.0\lambda \times 30.0\lambda \times 15.0\lambda$ (i.e., 4.29 km by 8.57 km by 4.29 km) domain with an added 0.28 km PML layer to absorb outgoing waves on the sides and bottom. A Ricker wavelet source was added at the coordinate (2.14 km, 0.43 km, 2.14 km) and 216 point receivers were arranged in a cubic grid with width of $5\lambda$ (i.e., 1.43 km) that was placed at a center offset of $10\lambda$ (i.e., 2.86 km) to

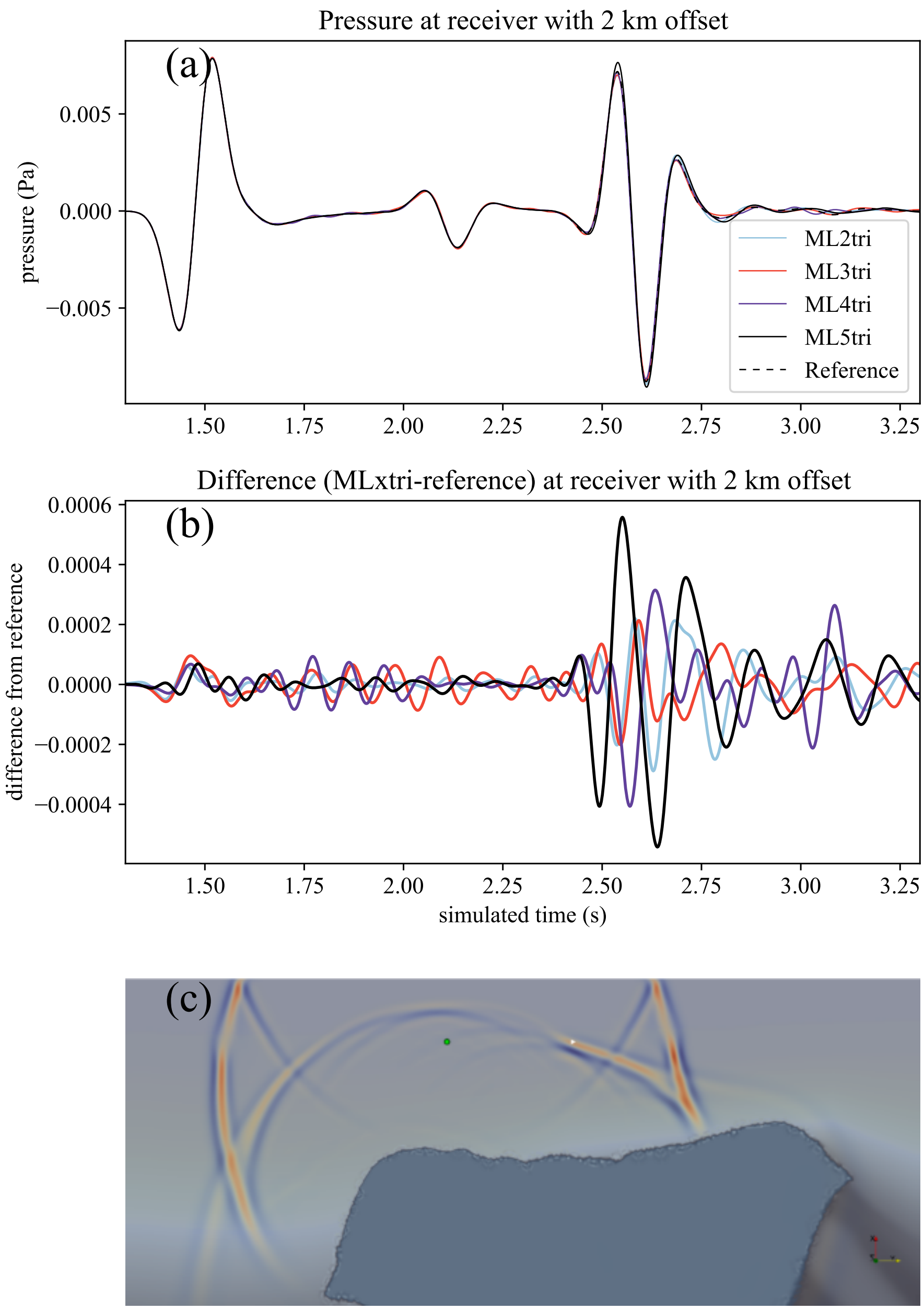

**Figure 11.** Panel (a) shows time series of pressure for several elements measured at a receiver with an offset of 2.00 km. The difference ($MLxtri - reference$) in signals from the reference case is shown in panel (b), where $x$ varies from 2 to 5. Panel (c) is the wave field at $t = 2.53$ s with the receiver at 2.00 km emphasized with a triangular glyph. The difference in the signals is greatest when the wave reflected by the salt body (indicated by darker blue in panel (c)) passes through the receiver also illustrated in panel (c).

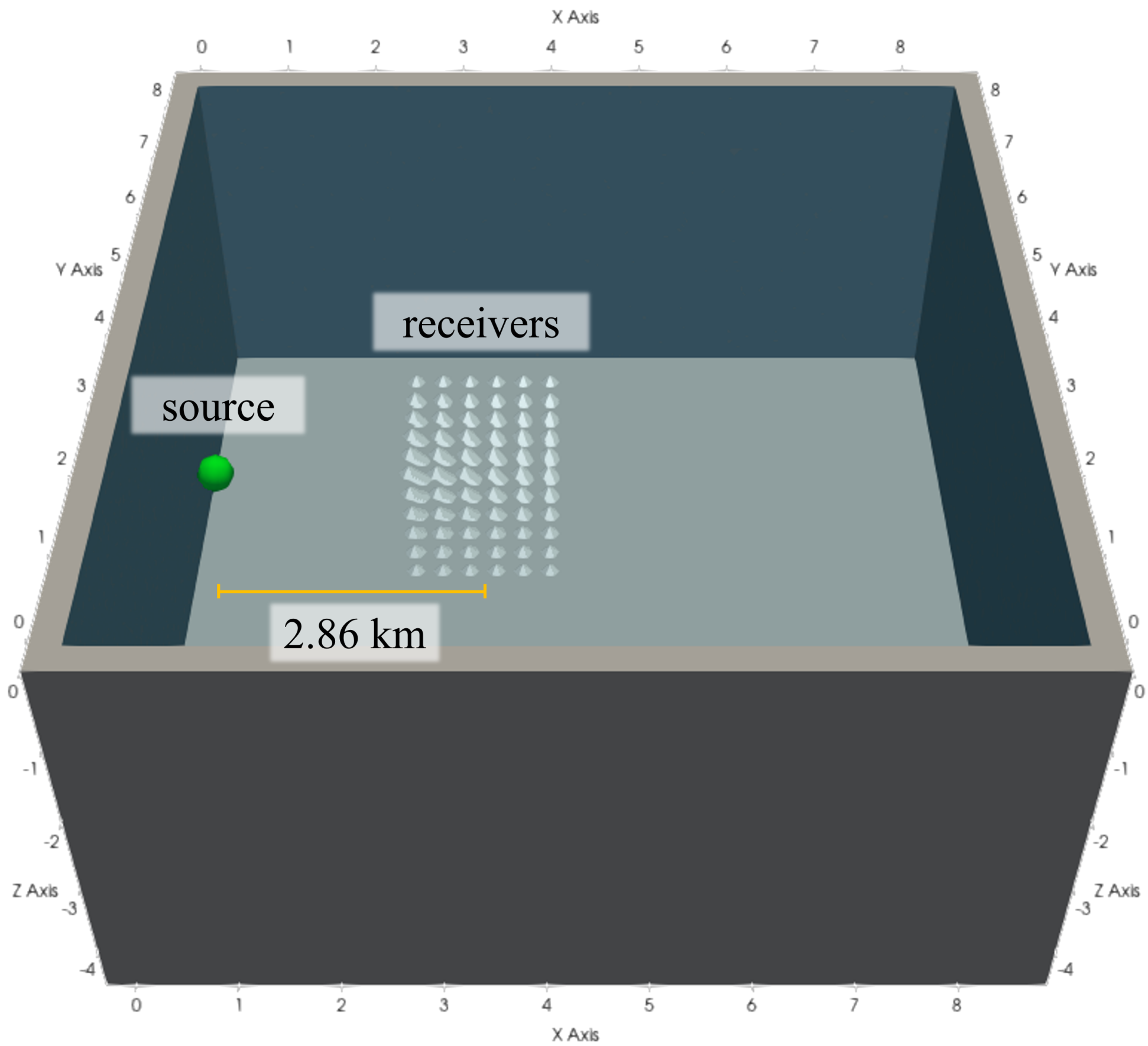

**Figure 12.** The 3D experimental configuration to calculate the grid-point-per-wavelength $G$ values. The Ricker source is represented as a green sphere and the receivers are denoted as white pyramid glyphs.

the right of the source coordinate, as illustrated in Figure 12. The timestep used for each simulation was $20.0\%$ less than the $\Delta t_{CFL}$ (maximum stable timestep based on an estimate) (c.f., Section 3.4). As with Subsection 5.2.2, meshes were generated by varying $C$ and a back-tracking line search was executed to reach an error threshold of $5.0\%$ calculated using Eq (40).

The results are shown in Figure 13. The $C$ values necessary to achieve $E = 5.0\%$ were $C = 5.1$ and $C = 3.1$ for $ML2tet$ and $ML3tet$, respectively. These results are similar in magnitude to the values found in $C$ for the 2D grid sweep for the $ML3tri$ of $C = 3.08$ but less than for $ML2tri$ which was $C = 5.85$.

### 5.3 Computational performance

Simulations were executed on a cluster called Mintrop at the University of São Paulo. Experiments used 4 Intel-based computer nodes. Each Intel node was a dual socket Intel Xeon Gold 6148 machine with 40 cores clocked at 2.4 GHz with 192 GB of RAM. Nodes were interconnected together with an 100 Gb/s InfiniBand network. While each node contained 40 cores, only

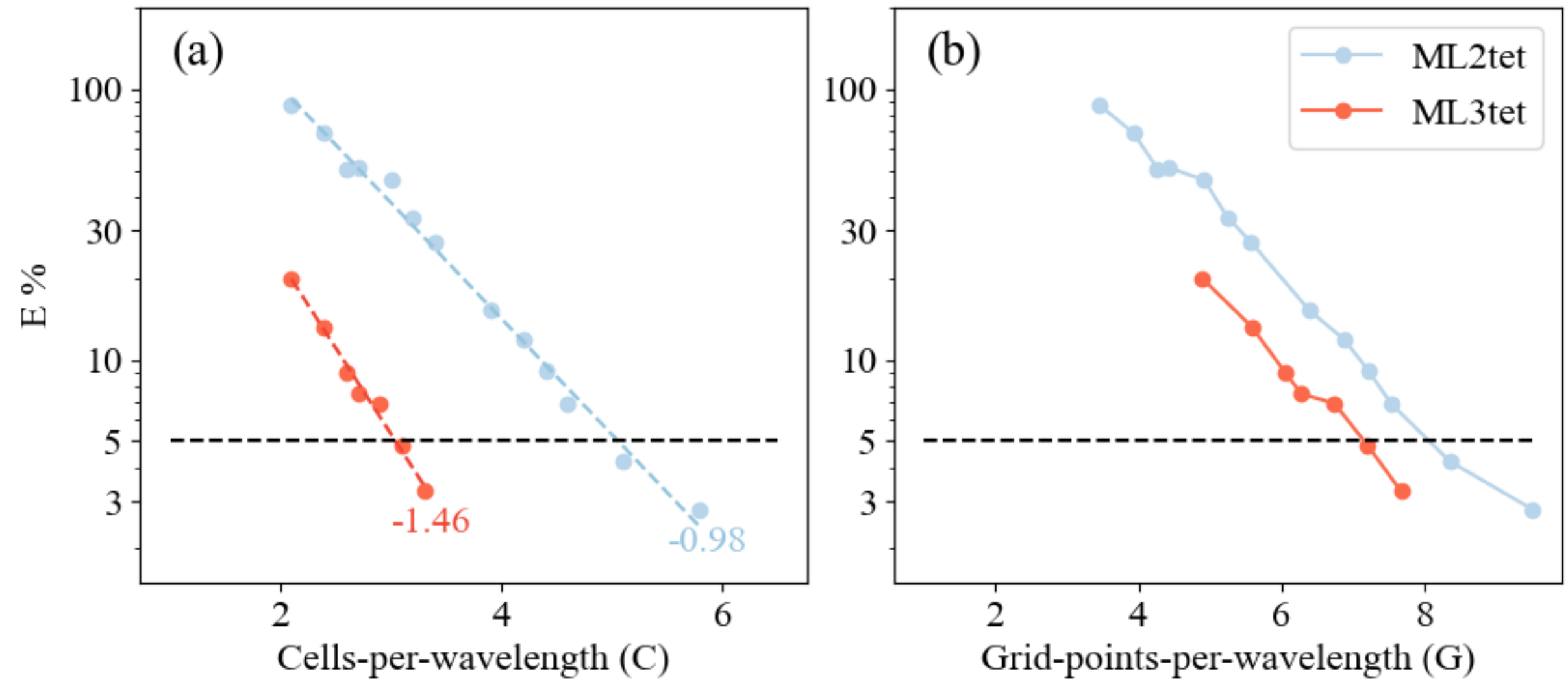

**Figure 13.** The 3D grid sweep, similar to Figure 8 but for 3D elements. The $E = 5\%$ threshold is drawn as a horizontal dashed black line on all panels. On panel (a), the line of best fit is drawn in the corresponding color and the slope of this line is annotated.

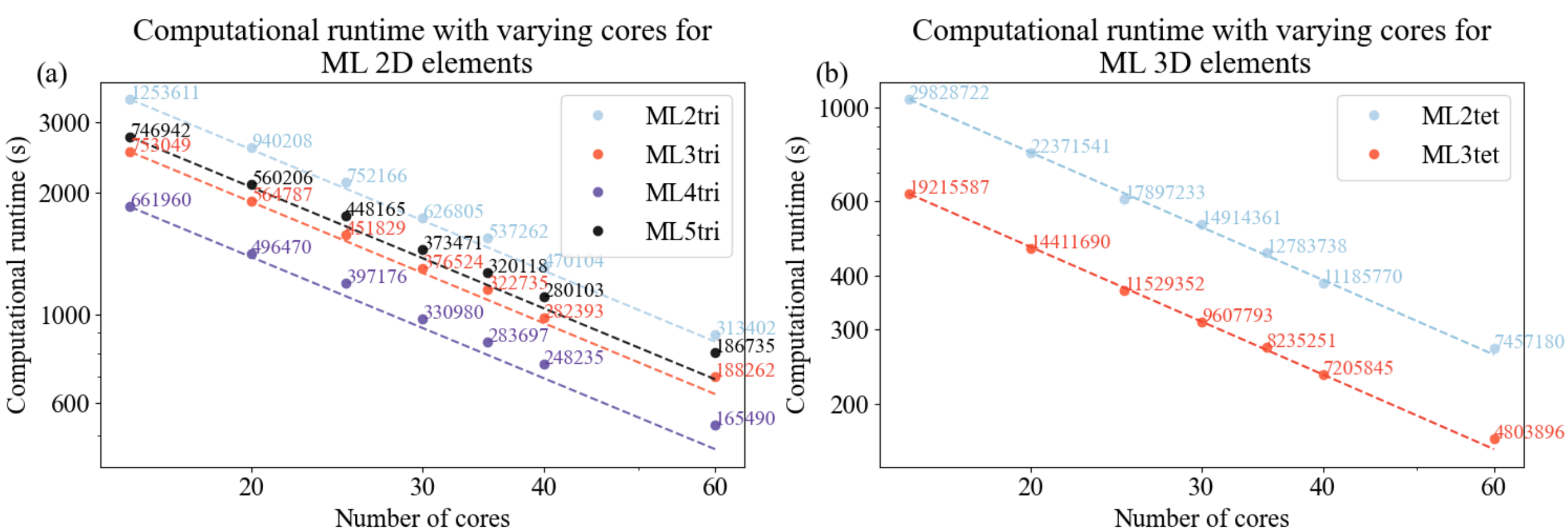

**Figure 14.** Strong scaling curves for solving the acoustic wave equation with a PML in spyro for 2D (a) and 3D (b) cases given a range of computational resources using Intel nodes. The dashed lines represent ideal scaling for each element and the average number of degrees-of-freedom per core is annotated.

a maximum of 15 cores were used per node to minimize the effects of memory bandwidth on the performance of the wave propagation solves.

The parallel efficiency of our forward propagator was assessed in Intel-based CPUs see Figure 14. For the 2D benchmark, the domain contains a uniform velocity of 1.43 km/s and spans a physical space of 114 km by 85 km. The 2D domain was discretized using the homogeneous cell densities from Table 2 resulting in $18,804,171$, $11,295,747$, $9,929,409$, and $11,204,136$ DoF for $ML2tri$, $ML3tri$, $ML4tri$, and $ML5tri$, respectively. In addition to the physical domain, a 0.287 km wide PML was included on all sides of the domain except the free surface. A source term with a time varying Ricker wavelet that had a central frequency of 5.0 Hz was injected into the domain and a line of 15 receivers with offset varying from 2.0 km to 10.0 km recorded the solution. The 2D simulations were executed for 4.0 seconds with timestep of 0.5 ms.

The 3D domain had 8 km by 8 km by 8 km with an additional 0.287 km wide PML included on all sides of the domain except the free surface. The domain was discretized using cell densities calculated in Section 5.2.4 resulting in $447,430,835$ and $288,233,805$ for $ML2tet$ and $ML3tet$, respectively. A source term with a time varying Ricker wavelet that had a central frequency of 5.0 Hz was injected into the domain and a cubic grid of 216 receivers was placed with a 2.86 km offset. The 3D simulations were executed for 1.0 second with a timestep of 0.5 ms.

Overall, nearly ideal strong scaling was observed in both 2D and 3D cases for most of the elements tested up to 60 computational cores. Since the ML elements admit diagonal mass matrices that avoid the need to solve a linear system, additional MPI communication is circumvented, which greatly improves parallel scalability. We point out that this analysis considers the gridpoint-per-wavelength results when designing the mesh sizes and thus represents a practical workload configuration. Weak scaling is also observed out to average of $165,490$ DoF in 2D and $4,803,896$ DoF using 60 cores. With that said in 2D, scaling deviates somewhat from the ideal curve for $ML4tri$ between 40 and 60 cores. With 60 cores the $ML4tri$ features the smallest problem in terms of average number of DoF per core and symbolic operations can begin to inhibit parallel scalability. It is important to note that similar parallel performance was also obtained for the adjoint-state wave propagator as it is highly similar in operations to the forward-state propagator.

### 5.4 Experiment with Marmousi2

To investigate FWI (Algorithm 1) with variable unstructured meshes, several 2D inversions were performed using the Marmousi2 model (Martin et al., 2005) (Figure 15). The objective of this experiment was to intercompare the performance of FWI in terms of wall-clock time, peak memory usage, and final inverted model. All inversions used meshes with variable elemental resolution based on the results with the homogeneous velocity model detailed in Section 5.2.2. The Firedrake programming environment enables us to flexibly select the variable space order at run-time.

FWIs commenced from an initial P-wavespeed model obtained by smoothing the ground truth Marmousi2 model with a Gaussian blur that had a standard deviation of 100 grid points (Figure 15(a-b)). The water layer (i.e., region of the velocity model with P-wavespeed $< 1.51$ km/s) was made exact in the initial seismic velocity model and was fixed throughout the inversion process by setting the gradient to zero there.

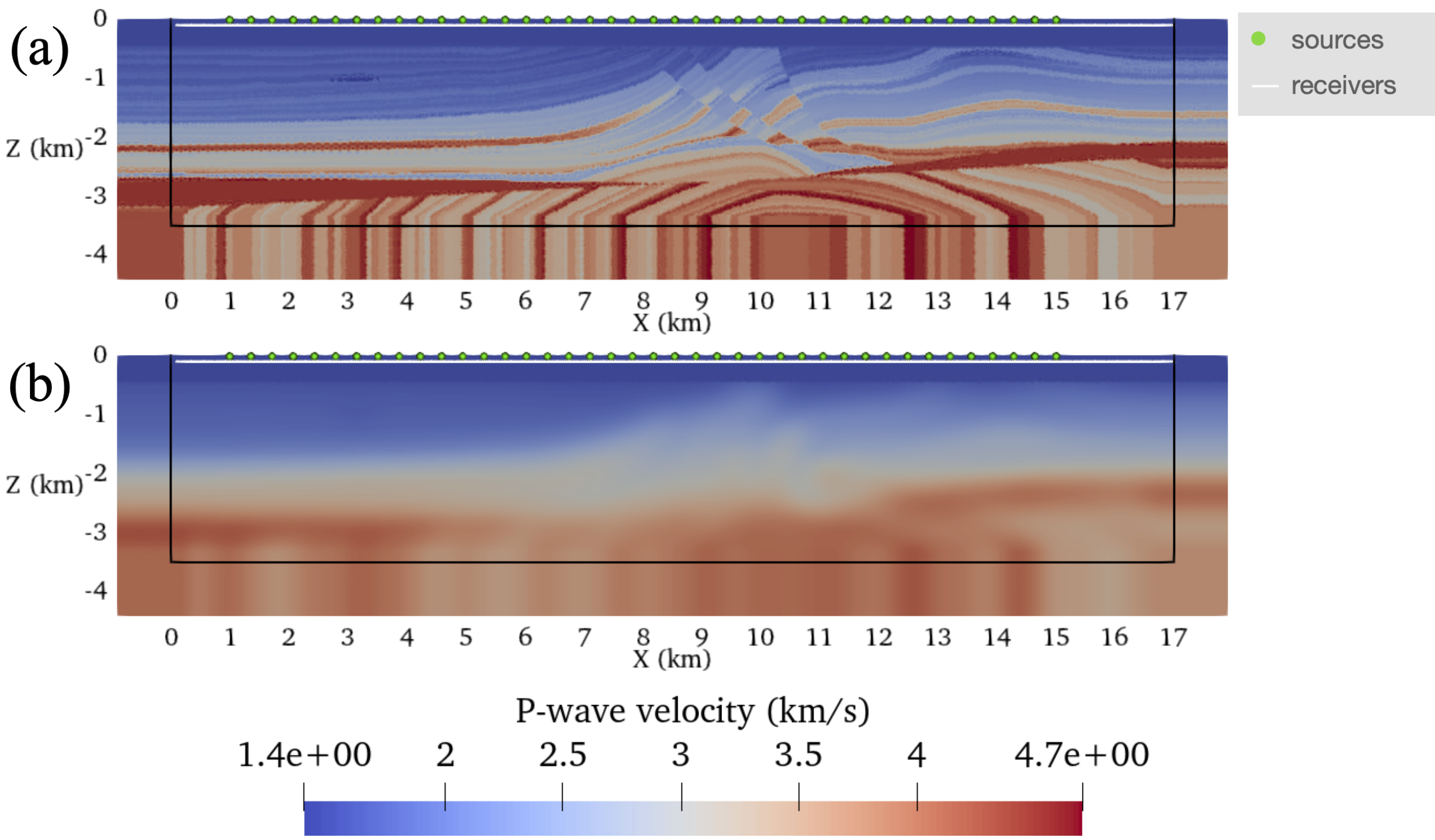

**Figure 15.** The Marmousi2 model setup described in Section 5.4. Panel (a) The target model, panel (b) the guess velocity model. On both panels, sources and receivers are annotated. The $\Omega_0$ is the region inside the solid-black line.

Each inversion used an acquisition geometry setup of 40 sources equispaced in the water layer between the coordinates $(-0.01, 1.0)$ km and $(-0.01, 15.0)$ km. A horizontal line of 500 receivers were placed at 100.0 m deep below the water layer between $(-0.10, 0.10)$ km and $(-0.10, 17.0)$ km. Simulations were integrated for 5.0 seconds with a noiseless Ricker wavelet that had a peak frequency of 5 Hz. A PML was added to the domain with a width $c_{max}/f_{max} = 900$ m (Kaltenbacher et al., 2013) and the non-reflective boundary was used to suppress free-surface multiplies (Eq. (6)).

The FWI setup described in Algorithm 1 was run for a maximum of 100 iterations $iter_{max} = 100$. Note that an iteration is only counted if it reduces the cost functional. The inversion process was terminated if either a) the norm of $\mathcal{G}$ was less than $1e^{-10}$ or b) a maximum of 5 line searches were unable to reduce $J$. However, neither criteria was reached in this experiment. A lower bound on the control $c$ of 1.0 km/s and an upper bound of 5.0 km/s were enforced throughout the optimization to ensure the result remained physical. Simulations were executed in serial using a numerical stable timestep of 0.001 seconds with a subsampling ratio $r = 10$, which yields a gradient calculation frequency 10 times less than the Nyquist frequency as determined by the 5 Hz peak source frequency.

Except for the $ML1tri$ experiment, all meshes for the initial velocity model were generated using the $C$ from Table 2 with an additional 20% to take into account the heterogeneous velocity model of Marmousi2 Table 3. It is general practice to increase

| Element | # DoF | $C$ | Run time (minutes) |
|---|---|---|---|
| $ML1tri$ | $139,605$ | 20.0 | 505 |
| $ML2tri$ | $103,877$ | 7.02 | 647 |
| $ML3tri$ | $71,561$ | 3.96 | 572 |
| $ML4tri$ | $54,592$ | 2.67 | 472 |
| $ML5tri$ | $56,995$ | 2.03 | 564 |

**Table 3.** The number of degrees-of-freedom (DoF) for each experiment, the cells-per-wavelength $C$ used to generate the mesh, and the total wall-clock time to run each FWI discretized with a different element type.

the $C$ for heterogeneous velocity models (Lyu et al., 2020; Anquez et al., 2019). In the case of $ML1tri$, the only possible mesh configuration that was capable to maintain the threshold error below 30% was $C = 20$. The so-called ground truth shot records that were used to drive the inversion process were simulated with a separate mesh discretized using the ground truth velocity model (c.f., Figure 15(a)) with $ML5tri$ elements using $C = 2.03$. Ground-truth simulated used a smaller timestep than what was used in FWI of 2.5 ms to minimize error associated with the time discretization.

The simulations were performed using one-shot-per-core using the shot-level ensemble parallelism described in Section 5.3 with 40 computational cores of one Intel node. Throughout each inversion, the total Random Access Memory (RAM) as a function of iteration, the total wall clock time spent performing the inversion, the cost functional $J$ (Eq. (10)) at each iteration, and the total number of iterations were recorded and documented.

### 5.4.1 Results

The number of DoF varied by approximately a factor of two over the range of ML elements tested. As expected, the $ML1tri$ produced the largest problem size with $139,605$ DoF whereas $ML4tri$ produces the smallest problem size with $54,592$ DoF. Note that all discretizations used a $\Delta C = 20.0\%$ (Eq. (41)) to take into account heterogeneity in the velocity model. It is interesting to point out that $ML5tri$ had a greater number of DoF in the problem than $ML4tri$ despite containing both higher-order basis functions and a lower $C$. We also note that in spite of going up to $ML5tri$, the variable mesh resolution enabled all FWIs to be simulated at a 1 ms timestep.

The final inverted models are shown in Figure 16 and are qualitatively highly similar to each other. Given that all forward discretizations were constructed with the same tolerance for $E$, this is to be expected. All experiments exhibited between 6 and 11 failed line searches during the course of the 100 iterations demonstrating no clear dependence between the number of failed line searches and the element type. With the exception of $ML1tri$, all results converged to a similar final cost functional between $4.88e10^{-3}$ and $5.21e10^{-3}$ after exhausting the iteration set. As compared to the other FWIs, the final cost functional for $ML1tri$ was largely greater by an order of magnitude ($J = 4.59e10^{-2}$), but still the inverted velocity model for $ML1tri$ qualitatively resembled the true velocity model.

The total run time memory and wall-clock varied substantially (Figure 17, Table 3). For example, $ML4tri$ produced the fastest FWI result completing in 472 minutes whereas in comparison $ML2tri$ produced the slowest result of 647 minutes. There was also a marked increase in total wall-clock time going from $ML1tri$ to $ML2tri$. Wall clock runtimes are primarily a result of right-hand side assembly time since solving the linear system with ML elements is pointwise division. Furthermore, the higher $k$ degree results in more shared nodes per element leading to more memory access and slower performance per DoF, which offsets the performance gains from reducing the problem size with variable mesh resolution. In regard to virtual memory usage however, there was a clear reduction in the peak random access memory (RAM) when ML elements were used, which was also noted in Lyu et al. (2020). For comparison, the $ML1tri$ element produced a peak RAM of 7.5 GB whereas $ML4tri$ required the least peak RAM of 3.1 GB. The $ML5tri$ required slightly more than $ML4tri$ with 3.13 GB.

### 5.5 Overthrust 3D section

As a demonstration of all the previous developments, the FWI implementation was applied to invert a section of the Overthrust3D P-wavespeed model (herein Overthrust3D) (Aminzadeh et al., 1996). Considering that the Overthrust3D is substantial in spatial extent (5.0 km deep x 20.0 m x 20.0 km), the focus of this section is to invert a still considerable 5.175 km by 7.5 km by 7.5 km section of the model (Figure 18(a-b)). The initial velocity model used to perform the inversion was obtained by smoothing the true velocity model using a Gaussian kernel with a standard deviation of 100 (Figure 18(b)). Similar to the other 2D FWI, the water layer (i.e., region of the velocity model with P-wavespeed $< 1.51$ km/s) was made exact in the guess velocity model and was fixed throughout the inversion process by setting the gradient in the water layer to zero. Finally, a 750 m PML is included on both true and guess models to absorb outgoing waves.

For the inversion, 20 sources were used that were laid out in a 2D grid composed of 5 lines equispaced along the $y$-axis with each line containing 4 shots equispaced along the $x$-axis (Figure 18(c)). All sources were located at the surface of the domain and the wave solution was recorded at a 2D grid of 900 receivers laid out 100 m below the surface. Each shot was simulated for 4.0 seconds, which was sufficient for the wave to spread out through the domain. A 5 Hz noiseless Ricker wavelet was injected at each source location.

Both the guess and true velocity models were discretized with $ML3tet$ elements. Each model featured elements adapted in size to the the true and guess model's local seismic velocity given a 5 Hz Ricker wavelet with a $C = 3.0$ that yielded $G = 6.97$ (c.f., Section 5.2.4). With this discretization, the guess problem contained 5.3M DoF whereas the true velocity model contained approximately 5.5M DoF.

Similar to the 2D FWI experiment, the 3D FWI ran for a maximum of 100 iterations $iter_{max} = 100$. The inversion process is terminated if either a) the norm of $\mathcal{G}$ was less than $1e10^{-10}$ or b) a maximum of 5 line searches were unable to reduce $J$; however, neither criteria was reached in this experiment. A lower bound on the control $c$ of 1 km/s and an upper bound of 6 km/s were enforced throughout the optimization to ensure the result remained physical. A numerical timestep of 0.75 ms was utilized and a gradient subsampling rate of $r = 20$ was used to conserve memory.

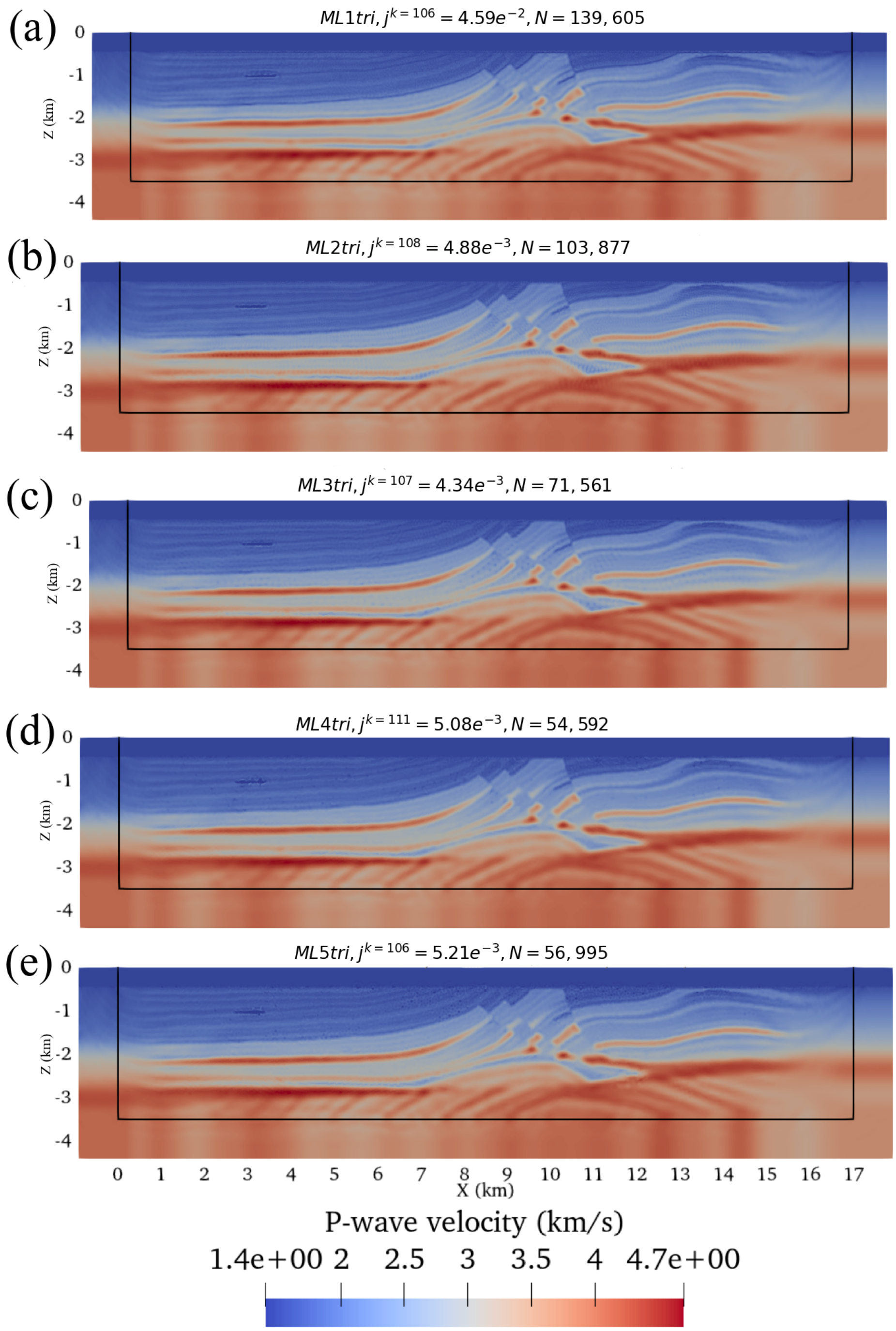

**Figure 16.** The final result for each FWI using different ML elements. The total number of iterations (including both iterations that reduced the cost functional and the ones that did not) are indicated in each figure along with the final $J$, and number of degrees-of-freedom $N$.

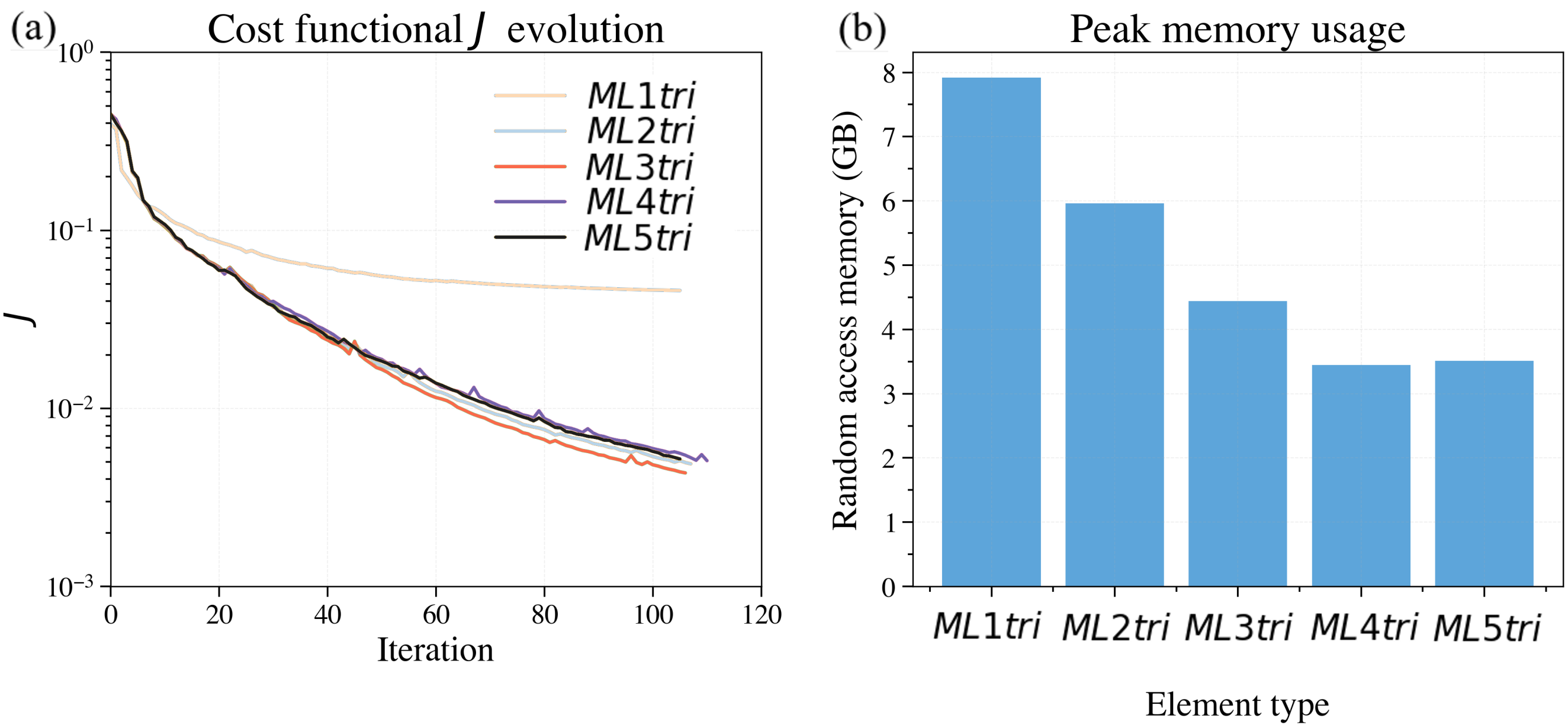

**Figure 17.** Comparing the performance of FWIs computed with different ML elements. Panel (a) shows the cost functional evolution and panel (b) shows the peak memory usage.

Simulations were performed using the two-level parallelism strategy with two AMD nodes. Each AMD-based node had an AMD EPYC 7601 machine with 64 cores clocked at 2.2 GHz with 512 GB of RAM. Specifically, each of the 20 shots used 6 cores for spatial parallelism requiring in total 120 computational cores.

#### 5.5.1 Results

The final inversion result along several cross-sectional slices along the $x$-axis and $y$-axis are compared with the true and guess velocity model (Figure 19, Figure 20). Overall, the inverted model demonstrates convergence to the true velocity model. After 100 FWI iterations, the cost functional reduced nearly one order of magnitude, from $4.76e10^{-1}$ to $6.62e10^{-2}$. Stratified layers appeared in the inverted velocity model that match structures and shapes in the true model, which are not present in the initial model. Overall, the inverted result appears more accurate near the surface closer to the sources than with depth, which is likely a result of poor source illumination beyond several kilometers of depth. Noise appears in the final inverted model however, which motivates the use of a regularization scheme in future FWIs.

Even with the use of mass-lumping elements and variable mesh resolution, 3D FWI remains computationally challenging on a relatively small-scale cluster with 120 cores. In this case, each FWI iteration took approximately 4.8 hours leading to a total continuous execution time of 20 days to perform 100 FWI iterations. Peak memory usage was significantly larger than in the 2D case at approximately 200 GB.

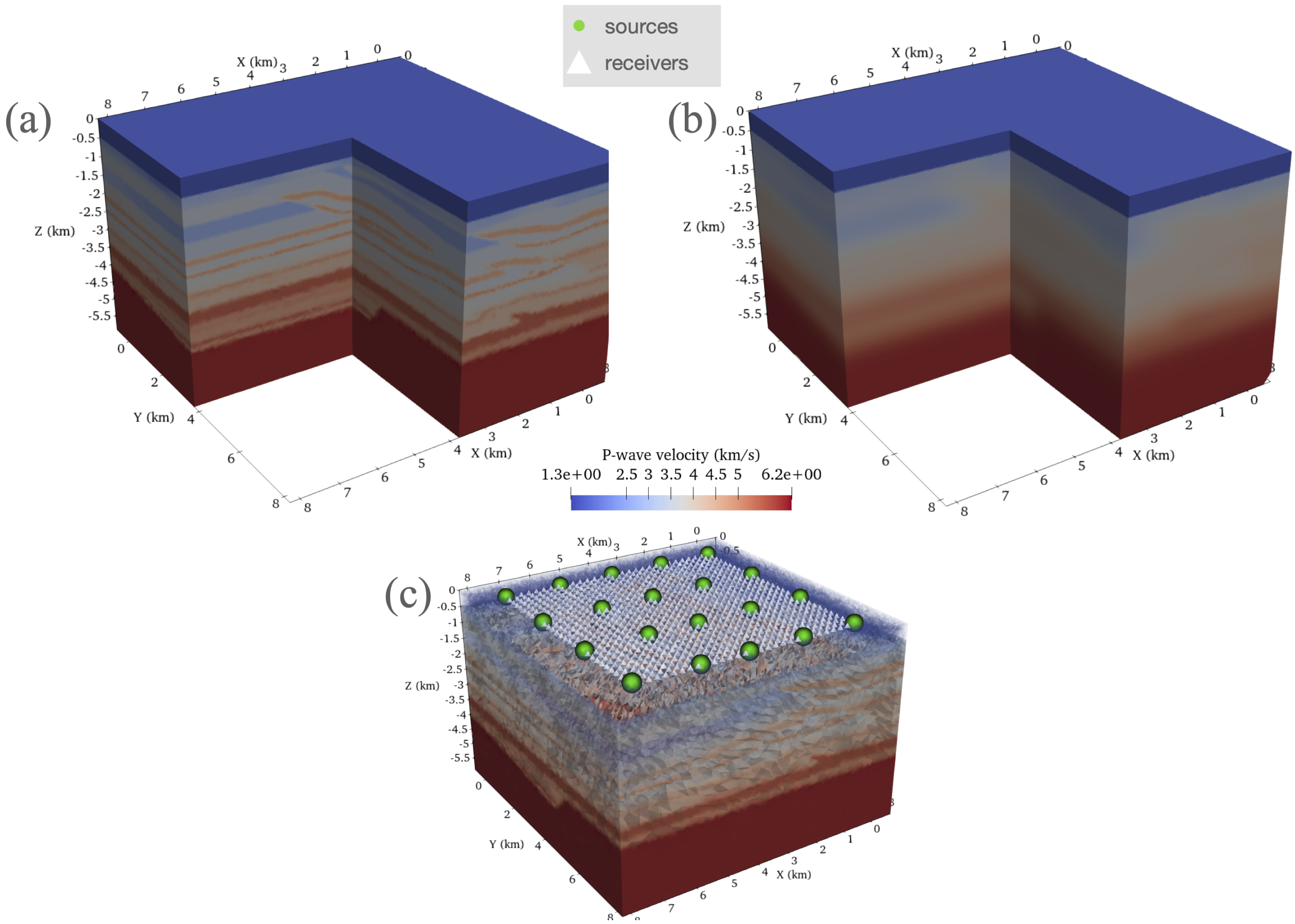

**Figure 18.** The Overthrust3d setup described in Section 5.5. Panel (a) shows the true model, and panel (b) shows the initial model. Panel (c) shows the location of sources and receivers.

## 6 Conclusions

We have discussed a methodology for imaging regional seismic velocity in two- and three-dimensional, arbitrarily heterogeneous, semi-infinite domains in a process commonly referred to as full waveform inversion (FWI). The FWI problem arises in geophysical exploration where high-fidelity imaging of seismic wave velocity is used to help locate oil and gas reservoirs. Acoustic waves are used to probe the domain, and the responses to the waves are recorded at an array of microphones near the surface. The FWI process involves solving a PDE-constrained optimization problem to minimize the misfit between the collected data and the computed response of the forward equations starting from some initial distribution of seismic velocity.

In order to resolve the wave equation, a continuous Galerkin (CG) finite element method (FEM) approach was developed using unstructured triangular (i.e., in 2D and tetrahedral in 3D) meshes with elements adapted in size to local seismic velocity.

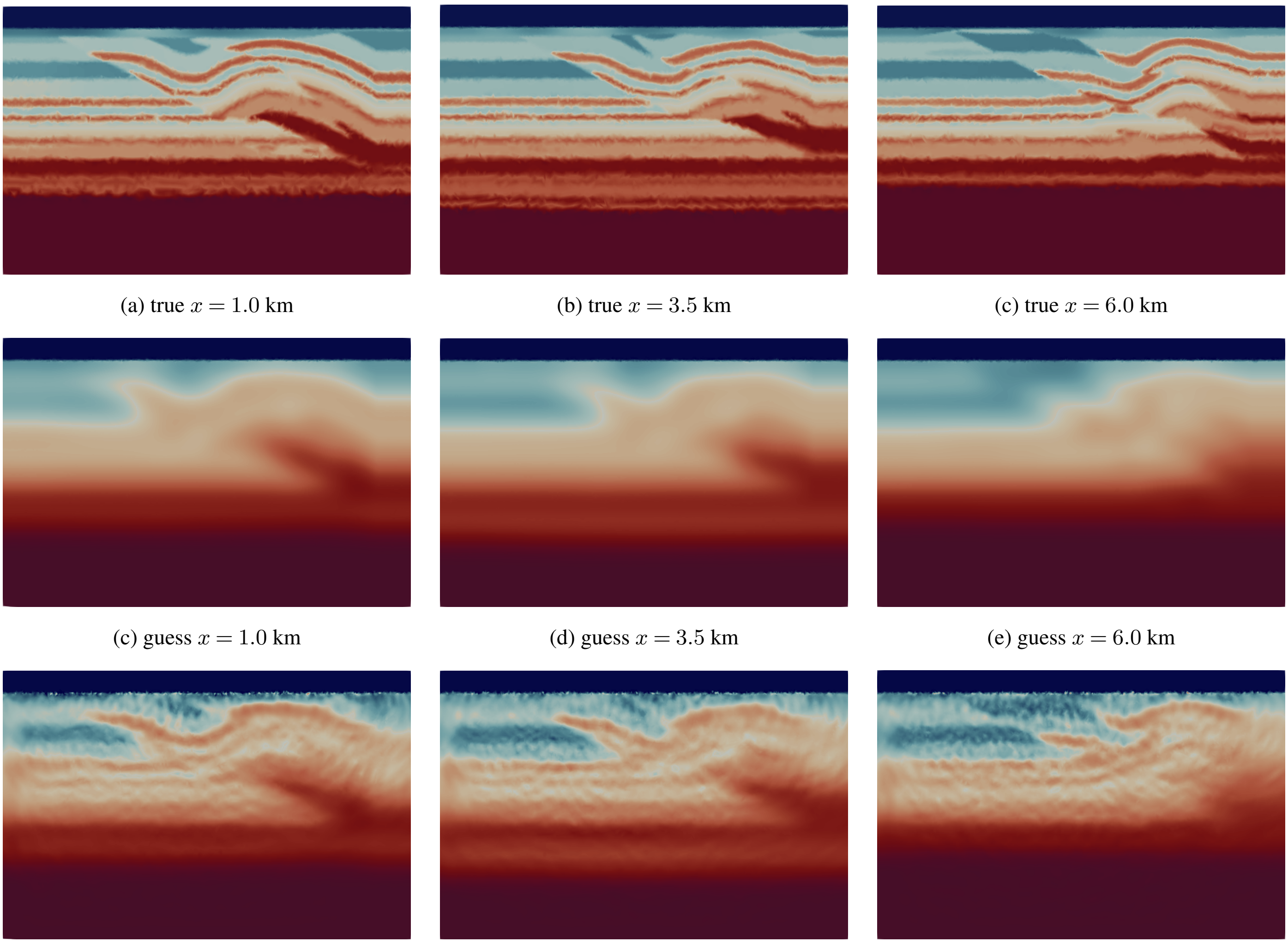

(a) true $x = 1.0$ km

(b) true $x = 3.5$ km

(c) true $x = 6.0$ km

(c) guess $x = 1.0$ km

(d) guess $x = 3.5$ km

(e) guess $x = 6.0$ km

(c) control $x = 1.0$ km

(d) control $x = 3.5$ km

(e) control $x = 6.0$ km

**Figure 19.** A comparison of cross-sectional slices along the $x$-axis in the Overthrust3D experiment between the true model, guess model, and reconstructed wavefield (control) after 100 FWI iterations.

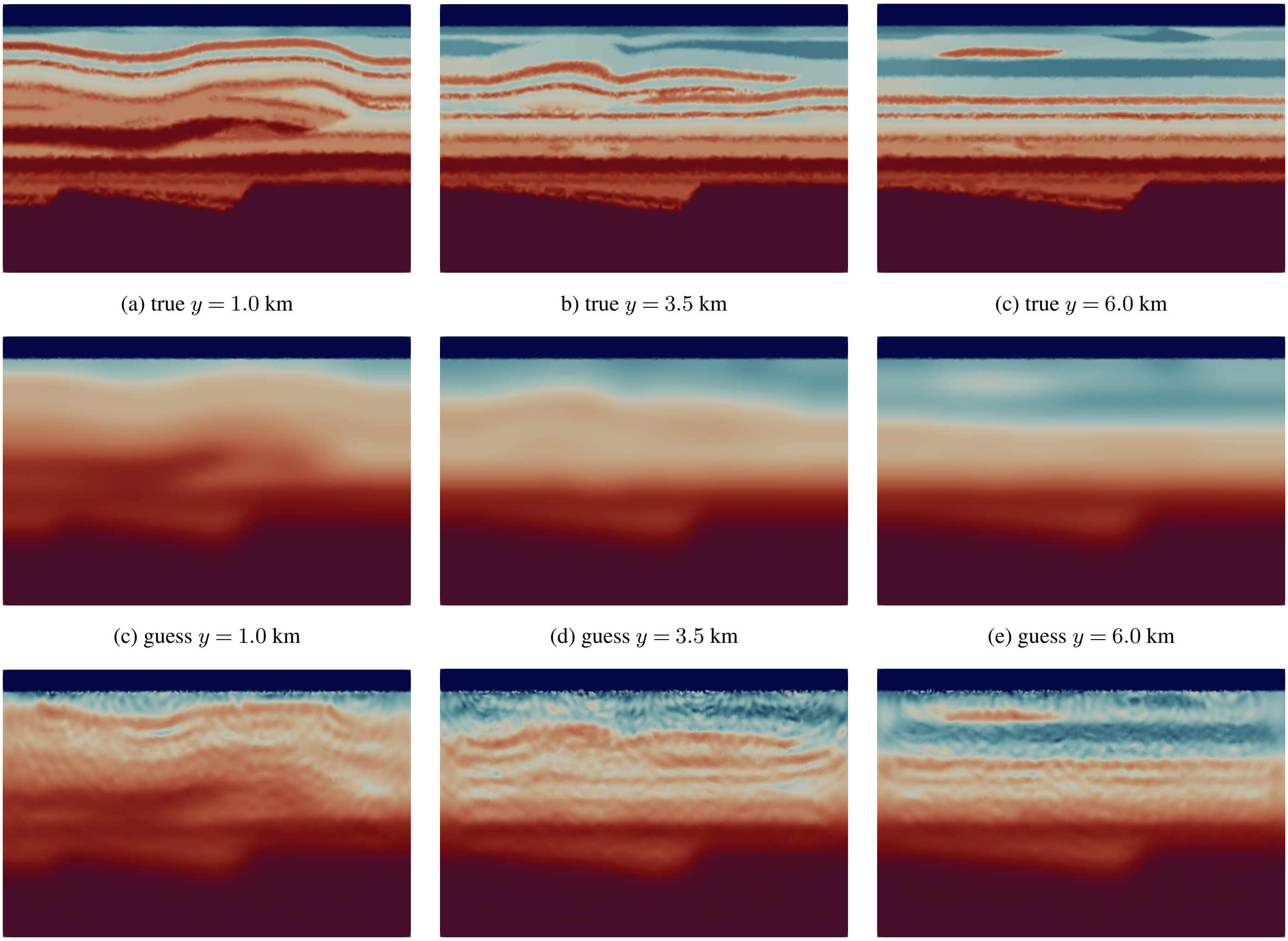

(a) true $y = 1.0$ km | b) true $y = 3.5$ km | (c) true $y = 6.0$ km

(c) guess $y = 1.0$ km | (d) guess $y = 3.5$ km | (e) guess $y = 6.0$ km

(c) control $y = 1.0$ km | (d) control $y = 3.5$ km | (e) control $y = 6.0$ km

**Figure 20.** Same as Figure 19 but for the $y$-axis

In this way, the design of mesh resolution becomes proportional to the wavelength of the acoustic wave hence the phrase waveform adapted meshes. Both the forward, the adjoint-state wavefields, and the gradient are computed on the same unstructured triangular mesh. To model a semi-infinite domain and suppress parasitic wave reflections, a Perfectly Matched Layer (PML) approach detailed in Kaltenbacher et al. (2013) was implemented. Both the discrete adjoint of the acoustic wave equation with the PML and its gradient with respect to the cost functional were derived using an optimize-then-discretize approach.

The FWI was implemented using the Firedrake package (Rathgeber et al., 2017) and the code is publicly available (https://zenodo.org/record/5164113). The Firedrake package enables us to represent the FEM discretization at a near-mathematical level simplifying our computer implementation. To solve the optimization problem, the Rapid Optimization Library (ROL; Cyr et al., 2017) was used and called directly from Python using pyROL (Wechsung and Richardson, 2019). Among other powerful features, ROL allows us to measure quantities in function space rather than Euclidean norms, which avoids well-known issues of mesh-dependent outer iteration counts for solving the optimization problems (e.g., Schwedes et al., 2017), which is not the case with some other common optimization packages (e.g., SciPy Virtanen et al., 2020).

An attractive aspect of the Firedrake package is the potential to utilize automatic differentiation (AD) (e.g., Dolfin-adjoint Mitusch et al., 2019) to derive the gradient from the forward discretization. At the time of development however, AD did not support the ability to annotate a solution at a point, which is necessary to define the cost functional in FWI applications. Future work intends to take advantage of AD as coding developments emerge. One could implement several forward wave solvers that each make different physical assumptions (e.g., variable density acoustic, elastic, visco-elastic, etc.) and discretize them in a common package using Firedrake. In this way, the user could readily control the physics and be able to solve more complex, multivariate FWIs without having to focus much effort on repeatedly deriving and implementing adjoint and gradient operators.

While triangular FEM offers flexibility to discretize domains with heterogeneous materials and irregular shapes, in the context of FWI they can lead to prohibitive computational costs to solve the sparse system of equations associated with their spatial discretization. As a result, five triangular 2D elements and three tetrahedral 3D elements that were originally detailed in Chin-Joe-Kong et al. (1999) and Geevers et al. (2018c) (referred to here as ML elements) that yield diagonal mass matrices (mass lumping) with special quadrature rules, were implemented inside the Finite Element Automated Tabulator (FIAT Kirby, 2004). Much like spectral elements on tensorial-based quadrilateral/hexaderal elements (Patera, 1984), ML elements were used to form a fully-explicit time marching scheme for wave propagation with a second order accurate in time scheme. We demonstrated that a 3D forward wave simulation could be scaled up in a distributed memory sense with close to ideal strong scalability. While SEM enables reduced-complexity sum-factored application of finite element operators in addition to lumped mass matrices, such techniques do not seem to be available for the enriched triangular elements that support mass-lumping. This issue would likely become significant at orders beyond those which are currently known for triangular elements.

In the context of FWI, the ML's mass lumping property also leads to performance benefits on the adjoint-state and gradient computations. Much like the forward-state calculation, the adjoint-state calculation is a wave propagation problem and benefits from mass lumping by avoiding solving linear systems of equations each timestep. Moreover, the diagonal mass matrix greatly accelerates the gradient calculation (19).

A major aspect of our FWI approach is that it takes advantage of variable triangular mesh resolution to discretize the domain by usingwaveform adapted meshes. Mesh resolution follows the local shortest wavelength of the seismic velocity model and is graded so that mesh resolution transitions are not too abrupt. We highlight that in order to successfully implement FWI with variable resolution meshes, automated (scripted) mesh generation tools are critically important (e.g., SeismicMesh Roberts et al., 2021). The mesh generation tool needs to produce high quality 2D/3D triangular meshes according to variations in local seismic velocity models, which reduces the overall number of DoF in the problem from that of a structured grid. To provide practical guidance for subsequent application in FWI, specific mesh resolution requirements were investigated to achieve a fixed error threshold of $5.0\%$ for 2D/3D forward wave propagation simulations using the ML elements. For a given fixed error threshold of $5.0\%$, in 2D the $ML3tri$ element achieved the target error threshold with the fewest number of DoF and in 3D the $ML3tet$ element achieved this goal. Linear elements could not meet any practical fixed error threshold and thus are not recommended for usage in FWI. Similar to the results in Lyu et al. (2020) for spectral elements, the usage of higher-order ML elements enabled us to greatly expand the element size while maintaining our desired accuracy. The expansion of the element size with higher-order elements has important implications with respect to generating 3D triangular meshes for regional and global seismic domains. In practical experience, the expansion of the element size can make the removal of degenerate sliver tetrahedral elements far easier, thus encouraging more numerically stable results with larger potentially numerically stable timesteps.

Higher-order ML elements of various orders led to similar final results in a synthetic 2D FWI. As the spatial order increased, we observed a small speedup to perform a fixed number of FWI iterations and a significant reduction in peak memory usage. On the down side, coarser meshes potentially under-resolve sharp velocity interfaces that could generate error in simulations. For forward wave propagation on heterogeneous velocity models, a homogenization technique would likely be necessary to simplify the velocity model before simulation is attempted to avoid complex mesh generation procedures. However in the context of FWI however, seismic velocity models are generally smooth and using very high degree elements has always been an option. In a similar fashion to the results outlined in Lyu et al. (2020) but using SEM, the primary benefit of higher-order ML elements was the ability to reduce peak run time memory requirements by nearly a factor of 2x in our 2D FWI example.

The work analyzed in this article presents several new directions for FWI with triangular FEM. In the course of the FWI process, the physical model incrementally evolves, and to aid convergence towards the global minimum of the cost function, a multi-scale reconstruction is often used by increasing step by step the frequency of the simulated phenomena. In the case of multiscale FWI, an automated meshing process in the FWI loop is then crucial to deal with the variations of the physical parameters and the increase of the frequency component of the waves simulated. Waveform adapted meshes could be used for each frequency of interest so as to obtain an accurate solution while using the coarsest mesh possible.

*Code and data availability.* All code used in this repository is free and open source, and all data sets used in the demonstrations are publicly available. The spyro source repository is available from https://github.com/krober10nd/spyro (last access: 20 August 2021). The spyro package is released under the GPLv3 license. The Zenodo release for the code is available at (https://zenodo.org/record/5164113), with data and

simulation scripts for FWI at (https://zenodo.org/record/5172307). This implementation was based on firedrake version (zenodo/Firedrake-20210810.0).

# Appendix A: Discretization details for the forward-state and adjoint-state equations

## A1 Expressions for the matrices

820 The expression of the matrices used in the forward discrete problem are:

$$\mathbb{M}_u = \mathbb{M}_\omega = \mathbb{M}_p^{x_k} = \int_\Omega \phi_i(\mathbf{x})\phi_j(\mathbf{x})d\mathbf{x} \quad \mathbb{M}_{u,1} = \int_\Omega \operatorname{tr}\Psi_1 \phi_i(\mathbf{x})\phi_j(\mathbf{x})d\mathbf{x} + \int_{\partial\Omega} c(\mathbf{x})\phi_i(\mathbf{x})\phi_j(\mathbf{x})ds$$

$$\mathbb{M}_{u,3} = \int_\Omega \operatorname{tr}\Psi_3 \phi_i(\mathbf{x})\phi_j(\mathbf{x})d\mathbf{x} \quad \mathbb{M}_{p,1}^{x_k,x_l} = \int_\Omega \psi_i(\mathbf{x})\Psi_1^{k,l}\psi_j(\mathbf{x})d\mathbf{x}$$

$$\mathbb{K} = \int_\Omega c^2(\mathbf{x})\nabla\phi_i(\mathbf{x}) \cdot \nabla\phi_j(\mathbf{x})d\mathbf{x} \quad \mathbb{D}^{x_k} = \int_\Omega \phi_i(\mathbf{x})\frac{\partial \phi_j}{\partial x_k}d\mathbf{x}$$

$$\mathbb{D}_{u,2} = \int_\Omega \psi_i(\mathbf{x})\Psi_2^{k,l}\frac{\partial \psi_j(\mathbf{x})}{\partial x_l}d\mathbf{x} \quad \mathbb{D}_{\omega,3} = \int_\Omega \psi_i(\mathbf{x})\Psi_3^{k,l}\frac{\partial \psi_j(\mathbf{x})}{\partial x_l}d\mathbf{x}$$

*Author contributions.* KR, AO, LF implemented the code, developed the numerical formulation, performed several results for the manuscript, and together wrote the manuscript. RK helped implement some aspects of the finite elements used, wrote and improved the manuscript. BS and RG provided the research environment and intellectual discussion necessary for the software's development and eventual realization of this paper.

825 *Competing interests.* The authors declare that they have no conflict of interest.

*Acknowledgements.* This research was carried out in association with the ongoing R&D project registered as ANP 20714-2, "Software technologies for modelling and inversion, with applications in seismic imaging" (University of São Paulo / Shell Brasil / ANP) – Desenvolvimento de técnicas numéricas e software para problemas de inversão com aplicações em processamento sísmico, sponsored by Shell Brasil under the ANP R&D levy as "Compromisso de Investimentos com Pesquisa e Desenvolvimento".

830 The fourth author (RCK) acknowledges support from the National Science Foundation grant 1912653. The sixth author (BSC) acknowledges financial support from the Brazilian National Council for Scientific and Technological Development (CNPq) in the form of a productivity grant (grant number 312951/2018-3).

We would like to thank Gerard Gorman at Imperial College London for his valuable feedback. We also acknowledge and appreciate the valuable feedback given from Wim Mulder, Amik St-Cyr, and Jorge Lopez from Royal Dutch Shell regarding full waveform inversion and finite elements.

We would also like to thank João Moreira for the generous comments and discussion.

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
