# Peer review of "spyro: a Firedrake-based wave propagation and full waveform inversion finite element solver"

_Geoscientific Model Development, 2021_

## Author Comment (AC1)

**Response to Anonymous Referee #1 (RC1)**

**spyro: a Firedrake-based wave propagation and full waveform inversion finite element solver**

Thank you for submitting your manuscript. The paper reads nicely and is well written. The objectives and proposed results are interesting and are a good fit for this journal.
Overall, I would recommend minor revision as some details need to be improved and the presented results are not fully convincing. I think this software could be of benefit to the geophysical community and that an improved manuscript would greatly improve its outreach.

R: We thank the referee for the time spent reading and evaluating the manuscript, and for the comments drawn. We are happy they see a valuable contribution in our work and appreciate their intention in promoting the improvement of the manuscript. Changes are annotated in the colour red and responses are annotated in the colour black.

Mainly my concerns are:

- The computational aspect, emphasised in the introduction, needs more clarity and some type of baseline

R: While we understand the point here, the computational aspect of FWI with this method is not the main focus of the manuscript. As we stated in the introduction, the aim of this paper is to provide solutions to the three enumerated issues regarding the application of triangular continuous Galerkin finite element methods to perform full waveform inversion, specifically using the higher order mass lumped elements.  However, we do understand the referee's point and elaborated more on this aspect while answering the question about computational runtime. In the response to comments regarding the computational aspect below, we also note changes to the manuscript that are relevant to this concern.

- The author does not discuss meshing when a "good smooth model" is not available. What would be the strategy if the mesh needs to be modified at each (couple) iteration.

R: In our case, because FWI is done using a local optimization approach, a good starting model is a prerequisite for the FWI process (Virieux and Operto 2009). We note that this was already stated in Section 2 when describing the FWI process.

There are a good number of publications focused on how to generate those starting models, which range from recent techniques such as dynamic time warping and convolutional neural networks (Yao and Wang 2022) and initial models based on global optimization approaches (Mojica, Kukreja, 2019) all the way through to the more industry standard reflection traveltime tomography (Woodward et al., 2008). We also add that constraints can be enforced in the mesh connectivity to capture material gradients if they are known a priori.

However, since the construction of an initial starting model is not the current focus of our research we use a smooth velocity model based on the application of a gaussian filter.

- The 3D results are not very accurate. While I understand these take a lot of resources and can not necessarily be re-run, justification on the accuracy is needed to justify the use of that software.

**R:** This concern is discussed in depth while responding to the referee's comments regarding the 3D overthrust section further below.

The author landscape needs to better justify why the software they propose improves the current software for FWI.

R:  Our software improves the current software landscape in a few ways, such as proving an extensible code stack for FWI, implementing software that is integrated with automatic mesh generation using simplices, software that provides tools that guide that mesh generation by providing relevant metrics and runs using higher-order mass-lumped elements which make FWI computationally feasible with continuous Galerkin finite element methods.

The use of the Unified Form Language (UFL Alnæs et al., 2014), a high-level mathematically consistent Domain Specific Language that closely resembles mathematical notation, inside our code, combined with Firedrake (Rathgeber et al., 2017) makes it a good candidate for an extensible and maintainable code stack for FWI. Its use of the Firedrake framework allows additional implementations, such as automatically derived adjoints (Farrel et al. 2020) and a variety of different spatial (such as higher-order mass-lumped triangular elements or quadrilateral spectral elements) and temporal discretizations in the same software.

We currently have other teams of researchers taking advantage of this aspect by testing the addition of sharp-interface imaging , spectral element methods on quadrilaterals, automatic adjoint formulations (already added in the latest version of the code), elastic wave solvers and different temporal discretizations. This extensibility is also required because new higher-order simplicial mass lumped elements can still be found in the future and Firedrake allows them to be added to spyro's wave solver.  A new paragraph was added to Section 6.
.
*"The current package enables developers to implement  forward wave solvers that each make different physical assumptions (e.g., variable density acoustic, elastic, visco-elastic, etc.) and discretize them using Firedrake within the current API. In this extensible environment, automatic differentiation (AD) (e.g., Dolfin-adjoint (Mitusch et al., 2019) can be used to derive the gradient directly from the forward discretization.  We envision future iterations of the package in which the user can readily control the physics and be able to solve more complex, multivariate FWIs without having to focus much effort on repeatedly deriving and implementing adjoint and gradient operators."*

Please see below for more details about the manuscript.

Line 18:
This is a bit terse, other methods exist (NMO, WEMVA, kirchoff, Depth/time extrapolation) and are widely used.

R: We kindly point out that this section is describing the timestepping methods used for solving the wave equation with a numerical technique so it's not immediately clear how those methods the reviewer mentions are relevant.

Line 20:

"However, the FWI problem is challenging to apply in practice since there exists a non-unique configuration of data that can best explain the observations. "

This is commonly described as cycle skipping. Please add references to the literature.

R: Several sentences were added into the introduction.

 Paragraph 1 of Section 1.
*"This is due to the nonconvex nature of the objective function usually employed in FWI, namely the L2-norm of the residuals between the recorded field data and the synthetic modelled data. A common manifestation of this nonconvexity is cycle skipping, which occurs when the phase match between the observed field data and modelled data is greater than half a wavelength, causing erroneous model updates in the optimization process (Yao et al., 2019)."*

Line 47:
One other problematic issue (the main one I would say) that isn't mentioned is the model dependence of the mesh. If the meshing becomes velocity/Unkown dependent then in theory, it should be added to the mathematical definition of the problem and the gradient of the mesh with respect to the velocity must be taken. This problem leads in some cases into elements wrapping onto themselves.
This should be discussed by the authors, and considering the mesh independent of the inverse problem needs to be justified.

R: In the cases run to produce the results for this paper, the mesh is static throughout the inversion process (it is generated based on the initial guess model and remains fixed), so there is no need to take into account mesh changes in the formulation. Also, the gradient, necessary for the optimization process, is mesh-independent, since it results from inverting the mass matrix on the right-hand-side "R" in equation 36.

The static nature of the mesh is emphasised in Section 3.3

*"We note that the mesh is built by adapting elements according to the initial velocity model and is static throughout the inversion process."*

Figure 1: Did you make that figure? This looks fairly familiar so please add a reference in case this is from somewhere else.

R: Another member of our research team who is not one of the authors of this paper, Ph.D. candidate João Baptista Dias Moreira, made that figure.

We credited it to him in the caption in the revised version of the paper.

Equation 1: This is not the conventional way to define the wave equation in an acoustic medium. Ignoring the PML for simplicity, the acoustic wave equation (d'Alember operator) is defined as
`diff(u, (t, 2)) - c^2 laplacian(u)`. including the spatially varying wave speed as `div(c ^ 2 grad(u))` is quite unconventional, what is the justification behind this formulation?

R: In the acoustic wave propagation in the subsurface, the wave passes through media with different wave speeds. In finite elements, the local approximate solutions are defined within an element, and in our high-order formulation the wave speed does not need to be constant inside the element. So, since the domain is not homogeneous in terms of the wave speed, it is necessary to use the equation in this form.

Equation 9: Following the point above, the expression for the sensitivity is quite unconventional as well. (usually is either `-2/c^3` or `1` depending on whether the wavespeed or squared slowness `1/c^2` is considered.

R: Indeed, the derivation of the adjoint and gradient follow from the definition of the forward problem, which, as you pointed out in the previous point, is not "usual".

Gradient subsampling: This is known to lead to artifacts and inexact gradients. What exact subsampling ratio is used and how much information is lost with that subsampling (maybe compare gradients with and without subsampling).

R: The reviewer is correct in that gradient subsampling leads to artifacts and inexact gradients. We believe that this focus is beyond the scope of the manuscript. Gradient subsampling is a common method to reduce the memory footprint of the FWI solver and this aspect is covered in other papers as a major focus area particularly in the context of "checkpointing", as alluded to in Section 3.6. We point out the r subsampling factor was noted in both FWI examples.

*" It is noted that this is known to lead to artefacts and inexact gradients and requires careful tuning to ensure feasible memory runtime requirements while balancing the accuracy of the gradient. The $r$ subsampling factor is noted in the results. These aspects were extensively studied in Louboutin et al. (2019)."*

Line 379: " were not ready to be used in our FWI code." So did you implement the adjoint by hand?

R: As stated in Section 3.5, the adjoint was implemented "by hand" by deriving the adjoint. A new appendix A1 is added detailing this derivation. Our team has evolved the software and we now have automatic differentiation working in the code. This leads to the same results we obtained with the adjoint equations implemented "by hand", unfortunately at higher computational costs due to annotation overhead. We have updated this sentence in the manuscript accordingly.

Line 380: Rapid Optimization Library. Any reason you did not choose a standard python library easier to use (i.e scipy) since, as you mentioned in the introduction, the bulk of the computing is the wave equation solves. Therefore "fast" optimization implementation doesn't really impact the overall performance.

R: ROL was preferred over Scipy's optimization library because we were working actively with the developers of the ROL interface. It was allowing for more advanced optimization methods as well. In addition, scipy does not support the specification of the L2 norm.

In Section 4.0 Computer implementation

*"ROL was preferred over Scipy's optimization library because the Firedrake development team is actively involved with the developers of the ROL interface. Further the usage of ROL allowed for more advanced optimization methods than SciPy offered."*

Line 386: "we exclusively rely on the second-order optimization method L-BFGS (". L-BFGS rely heavily on exact gradient information. Subsampling the gradient leads to an inexact gradient that doesn't satisfy l-bfgs requirement to guarantee better convergence.

R: The precision of the gradient was not severely damaged by subsampling. Indeed, we verified that the shape of the gradient remained essentially the same as without the subsampling. It is true however that, if the frequency of the subsampling is not high enough, the gradient would be poorer, typically when the wave, travelling one mesh element "h" and experiencing a wave speed "c" would be sampled with fewer than, say, five points, leading to a condition of the form "dt<5h/c".

In Section 4.0:

*"Given the gradient subsampling (c.f., Section~\ref{sec:gradient_downsampling}, the precision of the gradient was not severely damaged by subsampling. This aspect was verified that the shape of the gradient remained essentially the same as without the subsampling. It is true however that, if the frequency of the subsampling is not high enough, the gradient can be damaged, typically when the wave, travelling one mesh element "h" and experiencing a wave speed "c" would be sampled with fewer than, say, five points, leading to a condition of the form "dt<5h/c"."*

Line 451: "parallelism is trivial and handled by splitting the MPI communicator into groups of processes at initialization" Using MPI for separable task parallelism is known to be overkill

and prone to issues. Any reason more adapted task parallelism (Dask, spark, ...) was not used?

R: Dask was not used because we found MPI performed adequately and was already integrated within the Firedrake library. We do note a benefit from using both "shot-level" and domain decomposition parallelism especially in 3D amd this was previously shown in Figure 14 in Section 5.3.

A new sentence was added:

*We do note a significant benefit from using both "shot-level" and domain decomposition parallelism simultaneously, especially in 3D, which is later detailed in Section~\ref{sec:parallel_perf}.*

The computational runtime, while informative, does not say much about the performance of the proposed method without any reference. How long would it take to run FWI with FD with the same amount of resources? And how much memory?

R: Again we stress the major aim of the manuscript is to detail the spyro package and its capabilities, not an intercomparison between methods. We indeed provided run times for parallel execution of the forward problem given suitable mesh resolution in Section 5.3.

We stress again that the novelty of our approach is a software capable of solving FWI using unstructured triangular meshes waveform adapted to a priori knowledge. Everything in our approach is geared towards using automatically generated unstructured meshes with variable resolution. Therein lies the difficulty in comparing with finite difference methods, which cannot use unstructured meshes with higher order approximation. There is almost no reason to use finite elements in structured meshes, and this has been shown in a number of publications, dissertations and theses.

One could start a comparison using regular grids for FEM. However, if we only do this comparison, then we contend that we are not actually comparing our implementation with FD. We would be completely taking out the novelty of our approach and redoing something that has already been researched extensively. Therefore, any comparison, in the authors' point of view, should take into account the FEM software implementations that were needed to be done in generating code completely compatible with automatically generated waveform adapted simplicial meshes. Naturally some of those implementations/features increase computational runtime.

Having said this, there are already a few comparisons of mass-lumped elements with FD and other methods in the literature, but those are usually restricted to the forward wave solver. Zhebel et al. (2014) did an interesting comparison of 5 specific 2D and 3D cases with varying interior complexity and topologies and arrived at the conclusion that:

> *The results show that, for simple models like a cube with constant density and velocity, the finite-difference method outperforms the finite-element method by at least an order of magnitude. Outside the application area of rectangular meshes, i.e., for a model with interior complexity and topography well described by tetrahedra,*

> *however, finite-element methods are about two orders of magnitude faster than finite-difference methods, for a given accuracy. (Zhebel et al., 2014)*

However, everything those authors did not only are based on exclusively the forward propagation, but also they consider perfect knowledge of the interior complexity (aligning elements with faults) and topology. In a FWI problem this cannot and perhaps should not be done.

We give one more example of the influence of using unstructured meshes. We have run a comparison, in 2D and using spyro, between 4th order triangular mass-lumped elements and 4th order SEM (using quadrilaterals, Gauss-Lobatto-Legendre collocation points and under-integration), which is already commonly used in FWI. Two different cases were studied, one using a homogeneous velocity model and another with the Marmousi velocity model. In order to run both cases, the cells-per-wavelength parameter chosen for SEM was based on the same experiment illustrated in section 5.2, therefore generating the same error (calculated by eq. 40) on the homogeneous model and using the same cell percentage increase for the heterogeneous case.

The first case, with homogeneous velocity, was done using the same domain and model parameters as the strong scaling test shown in section 5.3 on 5 cores. The computational time necessary for the forward simulation was 906 s and 648 s for the mass-lumped triangles and SEM, respectively. However, even though the triangle-based simulation took more time to run, it had less degrees-of-freedom then the SEM-based one (9,929,409 vs 10,212,615). Since both utilises diagonal mass matrices, the longer runtime can be explained by the structured mesh used in quadrilaterals, which increases time spent updating right-hand side values, with 596 s and 356 s spent in this section for triangles and quadrilaterals, respectively. The rest of the simulation took roughly the same amount of time (310 s and 292 s).

However, since we use waveform adapted elements, which increase element sizes in sections of the mesh and therefore lower degrees-of-freedom, we did one more comparison. An unstructured waveform adapted mesh, with triangular mass-lumped 4th order elements, was compared with a structured mesh, with quadrilateral SEM on the Marmousi velocity model. This is done only to illustrate the gains of waveform adapting the elements, since SEM can also use unstructured meshes in 2D, even though in 3D automatically adapting hexahedral meshes is considerably more difficult then tetrahedra. Once again, for both mass-lumped triangles and SEM, the cells-per-wavelength parameter was calculated based on the same error for both discretizations using the methodology illustrated in section 5.2. The adapted mass-lumped triangles outperformed the structured quadrilaterals with runtimes of 327 s and 755 s. It is important to note that this comparison was only done to illustrate gains associated with unstructured elements and the authors know that SEM can also use unstructured elements and there are many techniques for semi-structured meshes and homogenization (Lyu et al., 2020) that can also be applied.

In section 1 (introduction):

"These elements have been compared with finite difference schemes and have favourable

results for the forward wave propagation when interior complexity and topography are present that can be adequately modelled with unstructured tetrahedra (Zhebel et al., 2014)."

**2D Marmousi**

The peak RAM usage seems a bit high. AS a comparison, as standard FD on this model (17km x 3 km discretized on a 20m grid) requires `151*851*1251*4/(1024^3) = .6Gb` of memory for a wavefield sampled at 4ms (`r=4` for your case) which is about 5x-10X lower than what your method seems to require. Can you elaborate on the memory needs of your method?

R: The memory requirements are different than what you would expect from a conventional solver written in ANSI C and/orFORTRAN because of the overhead associated with the Domain Specific Language and the way Python handles memory management via garbage collection. Ongoing improvements in the Firedrake library aim at reducing the memory overheads, but this is an active issue with using the Firedrake library.

Another important issue is that some available FD software have since launch been continuously incremented with memory and performance enhancing upgrades. Some of which can also be applied here and should be done when comparing different softwares. Two relevant examples are checkpointing for FWI and source-encoding in the forward shot propagation. The latter allows the use of propagating groups of shots simultaneously, creately reducing runtime and memory costs.

The runtimes need a reference as by themselves they do not provide any information.
R: Please see the answer to the comment before the previous.

**3D overthrust**
The 3D overthrust results aren't very convincing as they show a lot of artifacts and noise. I would consider trying to improve those. The author states "which is likely a result of poor source illumination beyond several kilometers of depth.". The illumination is usually corrected for with the hessian (or some diagonal approximation such as `1/norm(u_^2)`. Because you use l-BFGS you should have an approximation of the hessian that is quite accurate after that many iterations. Therefore this interpretation seems incorrect. You can see in related work (such as Witte et. al you refer to) that the deep part of the model can be recovered with a god initial model such as yours.
R: The reviewer is correct about the existence of a lot of artefacts and noise. The reason for this is not the "poor source illumination beyond several kilometers in depth" mentioned earlier, but the low number of sources used. Because of the current lack of source encoding in spyro, which is something presently being implemented by other researchers, our 3D model was limited to only 20 sources. Source-encoding would allow us to run many shots simultaneously, greatly reducing memory costs and runtime. Because of the low number of sources, a large source interval of 1750 m was used, causing aliasing artefacts in the final model. Witte et al. (2019) uses 9600 sources, although on the full 3D model (5.0 km deep x 20.0 m x 20.0 km), which results in roughly 200 m source interval. Based on Brenders and Pratt (2007), to not have those aliasing artefacts, source spacing would ideally be below $\lambda/2$,

but can also be increased up to 3λ/2, since the minimum velocity, outside of the water, in the Overthrust 3D model is 2178 m/s, this translates to using a maximum source spacing of 653 m (with the 5 Hz frequency) in order to reduce those aliasing artefacts. The ideal value of source spacing would be 218 m.

The original statement has been changed in the manuscript, nad the following sentence has been added:

"Another issue present in the final model are aliasing artefacts, caused by the large interval between sources of 1750m. Ideally this interval would be λ/2, but can go up to 3λ/2 (Brenders and Pratt, 2007), which for this model and source frequency is 653m."

C:

A 5Hz peak wavelet is unrealistic as it leads to <1Hz data not being feasible in real life. Low-frequency FWI is usually performed for frequencies >3Hz. This could be achieved with a higher frequency wavelet (ie 15 Hz) than band/lowpass your data to 3-8Hz for inversion.
R: The reviewer is correct with regards to field application of FWI employing higher frequencies. The study with lower frequencies (5 Hz) aims to test the limits of the proposed approach even with a lack of information generated by the absence of higher frequencies. Even with a significantly low frequency, the results obtained were good. If we employed higher frequencies, we would in principle be able to better define finer features and thin layers. This is being investigated in an ongoing work, in which we are developing a multiscale FWI technique using spyro.
C:

**References**:
Brenders, A. J. and Pratt, R. G.: Efficient waveform tomography for lithospheric imaging: implications for realistic, two-dimensional acquisition geometries and low-frequency data, Geophysical Journal International, 168, 152–170, https://doi.org/10.1111/j.1365-246X.2006.03096.x, 2007

Lyu, C., Capdeville, Y., and Zhao, L.: Efficiency of the spectral element method with very high polynomial degree to solve the elastic wave equation, Geophysics, 85, T33–T43, https://doi.org/10.1190/geo2019-0087.1, 2020.

Mojica, O. F. and Kukreja, N.: Towards automatically building starting models for full-waveform inversion using global optimization methods: A PSO approach via DEAP + Devito, in: SEG Technical Program Expanded Abstracts 2019, pp. 5174–5178, Society of Exploration Geophysicists, San Antonio, Texas, https://doi.org/10.1190/segam2019-3216316.1, 2019.

Virieux, J. and Operto, S.: An overview of full-waveform inversion in exploration geophysics, Geophysics, 74, WCC1–WCC26, https://doi.org/10.1190/1.3238367, 2009.

Woodward, M. J., Nichols, D., Zdraveva, O., Whitfield, P., and Johns, T.: A decade of tomography, GEOPHYSICS, 73, VE5–VE11, https://doi.org/10.1190/1.2969907, 2008

Yao, G., da Silva, N. V., Warner, M., Wu, D., and Yang, C.: Tackling cycle skipping in full-waveform inversion with intermediate data, Geophysics, 84, R411–R427, https://doi.org/10.1190/geo2018-0096.1, 2019.

Yao, J. and Wang, Y.: Building a full-waveform inversion starting model from wells with dynamic time warping and convolutional neural networks, GEOPHYSICS, 87, R223–R230, https://doi.org/10.1190/geo2021-0168.1, 2022

Zhebel, E., Minisini, S., Kononov, A., and Mulder, W. A.: A comparison of continuous mass-lumped finite elements with finite differences for 3-D wave propagation: A comparison of mass-lumped FEM with FD for 3D, Geophysical Prospecting, 62, 1111–1125, https://doi.org/10.1111/1365-2478.12138, 2014.

---

## Author Comment (AC2)

**Response to Anonymous Referee #2 (RC2)**

In this manuscript, the authors present spyro, a full waveform inversion finite element solver based on Firedrake. The solver features wavespeed-adapted triangular/tetrahedral meshes, a fully-explicit time stepping scheme based on a mass-lumping technique, and conforming elements of variable orders (up to degree 5 in 2D, and 3 in 3D).

The paper is well-written and fits the scope of this journal. The numerical results also look promising and the proposed solver seems to have several advantages over other FWI methods.

R: We thank the referee for these comments and for the time he/she spent evaluating our work.

I only have a few remarks/questions:

-p3, line 61: spectral triangular elements are known at least up to degree 9 (see, e.g., Mulder 2013: New triangular mass-lumped finite elements of degree six for wave propagation, Cui et al. 2017: High order mass-lumping finite elements on simplexes, and Liu et al. 2017: Higher-order triangular spectral element method with optimized cubature points for seismic wavefield modeling).

R: The referee is right, and we thank him/her for calling our attention to that. We have updated the sentence in the paper and added references to Cui et al. 2017 and Liu et al. 2017, which presented 9th order mass lumped triangular elements.

However, our numerical results already gave diminishing returns at high order, so increasing from degree 5 is unlikely to help our application in practice.

-p6, line 161: \sigma_i is not defined here. I suggest to give the definition of \sigma_i and \Psi_i here and explain that p_i and w only need to be computed in the boundary layer.

R: We have modified the text to "... and $\Psi_{i}$ are the damping matrices. These damping matrices are calculated using damping functions, which are referred to as $\sigma_{i}$. Note that $\Psi_{i}$, $p$ and $\omega$ only need to be calculated in the PML."

-Section 2.2: is there a reference for the derivation of the adjoint equations? Especially the boundary conditions given on page 8, line 207 need some explanation. The adjoint problem has 2 boundary conditions while the original problem had only 1.

R: The original problem has actually an extra boundary condition, namely "(n.p)=0". This condition was implemented through making an integration by parts on the term "-\nabla.p" on equation (1) in the FEM formulation and taking the remainder boundary term to be zero. This was done since no clear boundary condition on "p" was originally mentioned in the paper by

Kaltenbacher et al 2013, and this one did stabilise the scheme. Also, the adjoint equations, together with their boundary conditions were derived by us in this work, and no reference on them was found. A derivation of the adjoint is now added in Appendix A1.

-p9, line 230: definition of F is missing.

R: Indeed, this definition was missing. It has been corrected in the new version.

-Section 3.1: is the stiffness matrix assembled and stored or are the computations matrix-free?
R: In the UFL firerake, the "right-hand-side" vectors of a linear system where the stiffness matrices would appear (terms related to matrices $A_n$ in equation (26)), are computed at every iteration without the knowledge of an a priori stored stiffness matrix. So the computations are matrix-free.

-p12, top line: In 3D, alpha is computed taking the cube root?
R: Indeed. This has now been clarified in the new version.

-p12, definitions $A_{n+1}$, $A_n$, $A_{n-1}$: please double check these definitions.
        --In $A_n$ top-right: should be $M_{\omega, 1}$ instead of $M_{\omega}$.
        --In $A_n$ bottom-right: should be 0.
        --In $A_{n-1}$ second row: second and third term should be swapped.
        --In $A_{n-1}$ bottom-right: should have a minus sign.
R: Thank you for pointing out those definition errors. They all have been corrected.

-p14, equation 36: the definition of G is not really clear. What is its continuous counterpart? Also, the right-hand-side should be a vector. Please give a more precise definition.
R: Indeed, the way the equation is written, the rhs is a scalar, but, if evaluated for every test function "delta c", makes a vector. We made a change in the text to clarify this.

-p16, line 396: 32 nodes instead of 50.
R: The referee is correct. This has been corrected.

-p17, Algorithm 1: step 10 is done via L_BFGS and step 11 via ROL? Or are both used for both steps?
R: Indeed, the text was misleading and was not clearly related to the algorithm. It has been clarified now. Both the descent direction and the step lengths are part of the l-BFGS algorithm and are implemented in ROL.

-p17, line 441: equation (36) instead of (19)?
R: The referee was correct. We have modified the cross-reference.

-p18, Figure 5: This figure is rather unclear. Does gradient.py do steps 8+9 of Algorithm 1? This figure also contains several equations, whereas I would expect steps of an algorithm.
R: We have modified the figure. gradient.py does both 8 and 9 of Algorithm 1.

-Section 5: it seems that the z-coordinate is always given first. Please explicitly mention this somewhere.

R: Okay this is now mentioned.

R: We have removed the linear fits.

R: They have been double checked.

R: Thank you for the suggestion, this is now annotated.

-Section 6: with 2 full pages, the conclusions seem to be overly long. Please try to make it shorter and more concise. A summary of the results and corresponding conclusions should be the main focus. Ideally, this section has one or just a few clear takeaway messages.

R: We have made the conclusions significantly shorter and more concise, it is now less than half of its original length.

0-p41, Appendix: The definition of superscripts x_k/x_l are missing. Also, does there need to be a summation over l in the last two equations?

R: Thank you for pointing that out. This was corrected in the new version.